



# Validation of TROPOMI tropospheric NO₂ columns using dual-scan MAX-DOAS measurements in Uccle, Brussels

Ermioni Dimitropoulou[1], François Hendrick[1], Gaia Pinardi[1], Martina M. Friedrich[1], Alexis Merlaud[1], Frederik Tack[1], Helene De Longueville[2], Caroline Fayt[1], Christian Hermans[1], Quentin Laffineur[3], Frans
Fierens[4] and Michel Van Roozendael[1]

[1] Royal Belgian Institute for Space Aeronomy (BIRA-IASB), Brussels, 1180, Belgium
[2] Université Libre de Bruxelles (ULB), Brussels, Belgium
[3] Royal Meteorological Institute of Belgium, Uccle, 1180, Belgium
[4] IRCEL-CELINE, Brussels, Belgium

*Correspondence to*: Ermioni Dimitropoulou (ermioni.dimitropoulou@aeronomie.be)

**Abstract**

Ground-based Multi-AXis Differential Optical Absorption Spectroscopy (MAX-DOAS) measurements of aerosols and tropospheric nitrogen dioxide (NO₂) were carried out in Uccle (50.8° N, 4.35° E) Brussels, during one year from March 2018 until March 2019. The instrument was operated in both UV and visible (Vis) wavelength ranges in a dual-scan configuration
consisting of two sub-modes: (1) an elevation scan in a fixed viewing azimuthal direction (the so-called main azimuthal direction) pointing to the Northeast and (2) an azimuthal scan in a fixed low elevation angle (2°). By applying a vertical profile inversion algorithm in the main azimuthal direction and a parameterization technique in the other azimuthal directions, near-surface NO₂ concentrations (VMRs) and vertical column densities (VCDs) were retrieved in ten different azimuthal directions. The dual-scan MAX-DOAS dataset allows partly resolving the horizontal distribution of NO₂ around the measurement site
and studying its seasonal variations. Furthermore, we show that measuring the tropospheric NO₂ VCDs in different azimuthal directions improves the spatial colocation with measurements from the Sentinel-5 Precursor (S5P), leading to a reduction of the spread in validation results. By using NO₂ vertical profile information derived from the MAX-DOAS measurements, we also resolve a systematic underestimation in S5P NO₂ data due to the use of inadequate a-priori NO₂ profile shape data in the satellite retrieval.

# 1 Introduction

Nitrogen oxides (NOₓ=NO₂ + NO) play a critical role in tropospheric chemistry through many gas phase and multi-phase chemical reactions. Tropospheric nitrogen dioxide (NO₂) is an important anthropogenic pollutant emitted by combustion processes associated with traffic, industrial activity and domestic heating (Seinfeld and Pandis, 1998). In the boundary layer, its lifetime is short (typically a few hours close to the surface) and, therefore, concentrations vary rapidly in time and space
(Ehhalt et al., 1992). NO₂ has a direct health impact and is considered as a proxy for air pollution, as its high concentrations





are often associated with high concentrations of other pollutants such as tropospheric ozone ($O_3$) and aerosols (Crutzen, 1979). Depending on the $NO_x$, $O_3$ and Volatile Organic Compounds (VOCs) concentration levels in the troposphere, two regimes can be distinguished: (1) the $NO_x$-limited and (2) the VOCs-limited regime. In the $NO_x$-limited regime, an increase of $NO_x$ concentrations leads to an $O_3$ increase, while in the VOCs-limited ($NO_x$-saturated) regime, the tropospheric $O_3$ increases when the VOCs concentration levels increase (Seinfeld and Pandis, 1998). For these reasons, it is essential to monitor the global distribution of $NO_2$ to assess chemistry transport models as well as for estimating air quality trends over different regions of the world.

Since more than two decades, satellite nadir measurements of atmospheric backscattered sunlight in the UV-Vis range have provided daily global tropospheric column measurements of atmospheric $NO_2$, as well as a number of other species such as sulphur dioxide ($SO_2$), formaldehyde (HCHO), glyoxal (CHOCHO), etc. Those observations started in 1995 with the ERS-2 GOME (Global Ozone Monitoring Experiment) instrument (Burrows et al., 1999), followed in chronological order by ENVISAT-SCIAMACHY (SCanning Imaging Absorption spectroMeter for Atmospheric CHartographY) in 2002 (Bovensmann et al., 1999), AURA OMI (Ozone Monitoring Experiment) in 2004 (Levelt et al., 2006), MetOp-A/ GOME-2A in 2006 and MetOp-B/ GOME-2B in 2012 (Munro et al., 2016). It is worth mentioning that the pixel size of those instruments showed a significant reduction from 40 x 320 km² for GOME to 13 x 24 km² for OMI. More recently, the TROPOspheric Monitoring Instrument (TROPOMI) sensor launched on board of the Sentinel-5p Precursor (S5P) platform in October 2017 reached an even finer resolution of 7 x 3.5 km² and further improved to 3.5 x 5.5 km² from August 6, 2019 onward. S5P is the first mission of the EU Copernicus Program dedicated to atmospheric measurements with a high spatio-temporal resolution. The improved spatial resolution of TROPOMI offers new opportunities for air quality monitoring compared to previous satellite instruments, but it also introduces additional challenges concerning large data storage, processing capability, and the difficulty in validating satellite measurements at high spatial resolution.

For about two decades, the Multi-AXis Differential Optical Absorption Spectroscopy (MAX-DOAS) technique (Hönninger et al., 2004) has been widely used for retrieving the vertical and horizontal distribution of trace gases and aerosols in the troposphere (e.g. Sinreich et al., 2005; Ortega et al., 2015). MAX-DOAS instruments perform observations of scattered sunlight in the visible (Vis) and ultraviolet (UV) spectral ranges at multiple elevation angles towards the horizon, leading to an increased sensitivity to absorbers situated close to the surface (Hönninger et al., 2004). $NO_2$, HCHO, CHOCHO, $SO_2$, bromine monoxide (BrO), water vapor ($H_2O$), nitrous acid (HONO) and tropospheric $O_3$ are some of the tropospheric species that can be measured by MAX-DOAS spectrometers (e.g. Wittrock et al., 2004; Pinardi et al., 2008; Clémer et al., 2010; Hendrick et al., 2014; Irie et al., 2011, 2012; Pinardi et al., 2013; Sinreich et al., 2007; Wagner et al., 2011 and 2018; Wang et al., 2018). Because of their ability to retrieve vertical column and profile information, MAX-DOAS measurements provide an adequate source of reference data for the validation of satellite nadir trace gases measurements (Irie et al., 2008; Peters et al., 2012; Wang et al., 2017). In the case of $NO_2$, studies published so far (e.g. Celarier et al., 2008; Irie et al., 2012; Kramer et al., 2008; Ma et al., 2013; Pinardi et al., 2020) indicate that satellite sensors tend to underestimate tropospheric $NO_2$ columns especially over large cities.



The motivation of the present work is to investigate the use of multi-azimuthal MAX-DOAS measurements in Uccle (Belgium) to validate the TROPOMI tropospheric $NO_2$ column observations in urban conditions. This measurement site is located south of Brussels, at a distance of around 6 km from the city center and is therefore representative of moderate to high pollution levels where the $NO_2$ spatial distribution can be highly heterogeneous. One complete year (March 2018 – March 2019) of

MAX-DOAS measurements is used to quantify the agreement between the two datasets and its seasonal dependence, and to investigate to which extent the multi-azimuthal capability of the Uccle instrument can contribute to improve this agreement. The paper is organized as follows: in Section 2, the measurement site and the MAX-DOAS experimental set-up are described, followed by the DOAS analysis and the retrieval methodologies applied to the MAX-DOAS observations as well as their validation. Section 3 focuses on the tropospheric $NO_2$ measurements performed by TROPOMI. Thereafter, in Section 4, results

are separated in two main parts: the demonstration of the dual-scan MAX-DOAS retrievals and afterwards, their use for the TROPOMI/S5P validation. Finally, in Section 5, concluding remarks and perspectives are given.

## 2 MAX-DOAS measurements in Uccle and retrieval methods

### 2.1 Measurement site and experimental set-up

The MAX-DOAS instrument operated at BIRA-IASB (Koninklijk Belgisch Instituut voor Ruimte-Aeronomie – Institut royal

d'Aeronomie Spatiale de Belgique) is an improved version of the system described in Clémer et al. (2010). Developed to contribute to the CINDI-2 intercomparison campaign in Cabauw, in September 2016 (Kreher et al., 2019), it was subsequently installed on the rooftop of the Royal Meteorological Institute (RMI) in Uccle (50.8°N, 4.34°E; 125 m a.s.l) and, since January 2017, continuously operated at this location. Uccle is situated to the south of the Brussels-Capital Region, one of the most densely populated areas of Belgium. Frequently, the $NO_2$ concentration monitored by the network of telemetric air quality

stations from Bruxelles Environment/Leefmilieu Brussel (https://environnement.brussels/) exceeds the European standards. $NO_2$ columns are also among the highest in Europe as detected by satellite sensors like OMI (Huijnen et al., 2010).

The MAX-DOAS dual-scan instrument is composed of three main parts: (1) an optical head mounted on a sun-tracker, (2) a thermo-regulated box with two spectrometers (UV and Vis) and (3) the acquisition unit. Optical fibers connect the optical head with the two spectrometers.

The optical head is equipped with a filter wheel that allows switching between skylight and direct-sun measurements. The UV optical fiber consists of a 6 m x 1000 µm long monofiber attached to a 2 m bundle made of 51 fibers. Likewise, the Vis fiber consists of a 6 m x 800 µm long monofiber attached to a 2 m bundle of 37 fibers. Both monofibers are placed at the focal point of a telescope lens of 50 mm focal length, resulting in a field of view of 1° and 0.5° for the UV and Vis channels, respectively. The optical head is also equipped with a digital inclinometer to correct for eventual misalignment of the elevation scanner.

Installed indoor, the thermo-regulated box is equipped with Vis and UV grating spectrometers covering the wavelength ranges of 405 to 540 nm and 300 to 390 nm, respectively. The UV spectrometer is from Newport (model 74086) with a spectral



resolution of 0.4 nm. To block the visible light and to reduce the stray-light in the UV wavelength region, a band pass filter (U-340 Hoya) is used. The output of the UV spectrometer is connected to a back-illuminated UV-enhanced Charge Coupled Device (CCD) detector system (Princeton Instrument Pixis 2K). The Vis spectrometer from Horiba (model Micro HR) has a spectral resolution of 0.7 nm and is also mounted on a back-illuminated CCD system (Princeton Instrument Pixis 100). Both

CCD detectors are cooled at 235 K (using multi-stage Peltier system). The overall spectrometric unit is thermally stabilized to better than 1°C.

To control the data acquisition, two computers are used. One records spectra coming from the Vis spectrometer and controls the sun tracker while the second, synchronized with the first one, records the spectra from the UV spectrometer.

In order to measure in dual-scan (elevation + azimuthal) viewing mode, the initial operation mode (elevation scanning in one

azimuthal direction) was modified. From March 2018, the instrument was operated in two modes: (1) a vertical scan mode covering nine different elevation angles in one fixed (standard) azimuthal direction (35.5° with respect to the North in the eastward direction) and (2) a horizontal scan mode covering 9 different azimuthal directions at a fixed elevation angle of 2° above horizon (Fig. 1). As can be seen in Table 1 and Fig. 9, several configurations were tested in order to select the best combination of horizontal and vertical viewing directions, which is a trade-off between the acquisition time and the horizontal

representativeness. Since the $NO_2$ emission sources are located towards the North, more azimuthal directions are selected in this direction. With an integration time of 60 s for each measured spectrum, the total scan duration (azimuthal + elevation viewing modes) ranges between 20 to 30 minutes, depending on the configuration. Each azimuthal direction was quality-checked by performing horizon scans as during the CINDI-2 campaign (Donner et al., 2019) in order to ensure that obstacles, like trees and buildings, are not present in the different lines of sight.

**2.2 DOAS analysis**

The spectra measured in both sub-modes are analyzed using the QDOAS spectral fitting software developed at BIRA-IASB (Fayt et al., 2011) for the retrieval of atmospheric trace gas abundances in the UV, Vis and near infrared spectral ranges. The DOAS technique consists in a separation between narrow absorption features characteristic of molecular species and a spectral background resulting mainly from Mie and Rayleigh scattering and instrumental effects (Platt and Stutz, 2008). Its primary

product is the differential slant column density (dSCDs), which represents the light-path integrated trace gas concentration in a measured spectrum relative to the amount of the same absorber in a reference spectrum. In the present case, daily noon zenith spectra are used as reference.

$NO_2$ dSCDs are retrieved in both Vis and UV ranges according to settings defined during the CINDI-2 campaign (see Tables 2 and 3; as well as Kreher et al., 2019). For retrievals in the Vis spectral range, we use the 425 - 490 nm fitting interval, while

UV retrievals are performed in the 338-370 nm spectral range.



### 2.3 Retrieval methods

#### 2.3.1 Aerosol and OEM-based profile retrievals

Aerosol extinction coefficient and $NO_2$ vertical profiles are retrieved for each MAX-DOAS elevation scan in the main azimuthal direction by applying the Mexican MAX-DOAS Fit (MMF; Friedrich et al., 2019) inversion algorithm to the corresponding measured $O_4$ and $NO_2$ dSCDs. The inversion is performed in two successive steps. First an aerosol extinction profile is retrieved using $O_4$ measurements according to the principles described in Frieß et al. (2006). This aerosol profile is then used as an input for the radiative transfer calculations needed to invert the $NO_2$ vertical profile.

The MMF algorithm uses the Optimal Estimation Method (OEM, Rodgers, 2000) formalism, the VLIDORT (Spurr, 2006) version 2.7 radiative transfer model (RTM) as forward model, and a Levenberg-Marquardt (LM) iteration scheme.

MMF works in linear measurement space and logarithmic retrieval space. Further details about this algorithm can be found in Friedrich et al. (2019). MMF was one of the retrieval codes used during the CINDI-2 campaign (Tirpitz et al., 2019) and it also participated in the Round-Robin comparison of profiling algorithms as part of the Fiducial Reference Measurements for Ground-Based DOAS Air-Quality Observations (FRM$_4$DOAS) project (Frieß et al., 2019).

An important parameter in the OEM approach is the a priori profile. In the present study, exponentially-decreasing a priori profiles are used for both aerosols and $NO_2$, with a scaling height of 1 km and aerosol optical depth (AOD) and $NO_2$ vertical columns fixed to 0.18 and 9.15 x $10^{15}$ molec cm$^{-2}$, respectively. For the diagonal elements of the a priori covariance matrix, we use 50 % of the a priori profile, with a correlation length of 200 m for the non-diagonal elements (Clémer et al., 2010).

The pressure and temperature profiles are taken from the Air Force Geophysics Lab (AFGL) 1976 Standard Atmosphere (Anderson et al., 1986). The variation of the temperature profiles during one year of measurements is taken into account as an additional error on the profile retrieval. The retrieval altitude grid consists of 20 layers of 200 meters thickness between the surface and 4 km altitude. The surface albedo is set to 0.06 and the aerosol optical properties such as the single scattering albedo and the asymmetry parameter are taken from co-located AERONET measurements. Regarding the retrieval wavelengths, aerosol extinction vertical profiles are retrieved at 360 and 477 nm and the $NO_2$ vertical profiles at 360 and 460 nm.

Each retrieval is quality-checked based on three different criteria. First, the degrees of freedom (DOFs) should be larger than one. This ensures that the profile information comes mostly from the measurements and not from the a-priori profile. Second, the relative root mean square error (RMS) of the difference between measured and calculated differential slant column densities with respect to (wrt) the zenith spectrum of each scan should be smaller than 15 %. This excludes local minima. Third, the AODs should be smaller than 5 because of the high profile uncertainties on the trace gas retrieval in such conditions (Hendrick et al., 2014). The above-mentioned criteria are applied to the $NO_2$ and aerosol profile retrievals in the Vis and UV ranges.





The presence of aerosols and clouds in the atmosphere can strongly affect the MAX-DOAS trace gas retrieval (Frieß et al., 2006; Gielen et al., 2014; Wagner et al., 2004, 2014 ). In order to exclude MAX-DOAS measurements, strongly influenced by the presence of clouds, a cloud filtering approach is applied using a co-located thermal infrared pyrometer. The pyrometer determines the total cloud cover fraction based on the temperature data over a field of view of 6° (Gillotay et al., 2001).

Generally, the method is able to determine most cloudy conditions, with the exception of cirrus clouds. The total cloud-cover fraction is defined as the ratio between the observed cloudy solid angle elements and clear-sky elements. In the present study, only MAX-DOAS scans with a total cloud-cover fraction less than 0.8 (80 %) are selected for further analysis. The application of the above-mentioned upper limit allows the rejection of scans under total cloud conditions and does not reduce significantly the total amount of the accepted MAX-DOAS scans.

The uncertainties of the vertical profiles retrieved by MMF include three types of errors (Rodgers, 2000): (1) the smoothing error, which represents the difference between the retrieved and the true profile due to the vertical smoothing, (2) the noise error, which represents the uncertainty of arising from the dSCD measurement and (3) the error coming from the forward model. In Table 4, an overview of the main error sources on the $NO_2$ near-surface VMR and VCD retrievals is presented. For the Vis range, the smoothing error amounts to around 3 % of the $NO_2$ near-surface VMR and 8 % of the VCD and the noise

error about 2 % of the near-surface VMR and 2 % of the VCD. In the UV range, the smoothing error amounts to around 2 % of the $NO_2$ near-surface VMR and 9 % of the VCD and the noise error is about 2 % of the near-surface VMR and 3 % of the VCD. Despite the fact that the smoothing error seems to be small, it is the main error source in the profile retrieval (Rodgers, 2000). The use of a constructed covariance matrix results in the underestimation of the smoothing error, because daily variations of the a-priori profile are not taken into account. The uncertainty associated to the forward model parameters has

been estimated by modifying the input parameters, such as the single scattering albedo and the asymmetry parameter, in the RTM calculations and quantifying the impact on the $NO_2$ near-surface VMR and VCD. For the Vis range, it is up to 3 % of the near-surface VMR and 4 % of the VCD and in the UV range, the corresponding values are 2 % and 1 %, respectively.
We also consider the systematic uncertainty on the $NO_2$ cross-sections at the assumed fixed temperature of 294 K which is about 3 %  (Vandaele et al., 1998). Taking into account the temperature dependence (0.4 % K$^{-1}$, Takashima et al., 2012) and

assuming a mean temperature difference between winter and summer of 23 K, the total systematic error due to $NO_2$ cross-sections reaches a maximum of 9 %. Combining all the above-mentioned sources of error, the following uncertainties for $NO_2$ retrievals are estimated: 11 % and 13 % on $NO_2$ VMR and VCD in the Vis range, respectively, and 10 % and 14 % on $NO_2$ VMR and VCD in the UV, respectively.

### 2.3.2 Dual-scan MAX-DOAS retrieval strategy

The dual-scan MAX-DOAS retrieval strategy refers to the $NO_2$ near-surface VMR and VCD retrieval over one complete MAX-DOAS elevation and azimuthal scan by using (1) an OEM-based profile retrieval and (2) a parameterization approach. The parameterization approach used in this study is an adaptation of the one introduced in Sinreich et al. (2013). It consists in a conversion of $NO_2$ dSCDs measured at one low elevation angle (2°) to near-surface box-average mixing ratios.





If sufficient aerosols are present in the lower troposphere (boundary layer), the measured dSCD at two low elevation angles (in the present study, $1^o$ and $2^o$) are equal. In this case, the concentration of an absorber close to the surface (e.g. $NO_2$) can be considered as box-average near-surface concentration $\bar{c}$ and is related to the dSCD as follows (Sinreich et al., 2013):

$$dSCD = \bar{c} \, dL_{eff} \tag{1}$$

where, dSCD is the differential slant column density of the absorber in molec $cm^{-2}$ and $\bar{c}$ its mean concentration in molec $cm^{-3}$ along the differential effective path lengths ($dL_{eff}$) in cm.

The unknown variable in Eq. (1) is the differential effective path length of the measurement. The absorption of the oxygen collisional dimer ($O_4$) can be used as a tracer for the light path distribution (Wagner et al., 2004). The concentration of $O_4$ is proportional to the square of the concentration of molecular oxygen $O_2$ which can be accurately determined. Variations in the

$O_4$ dSCD are therefore directly related to changes in the state of the atmosphere and changes in the measurement geometry.

For each measurement, the differential effective path lengths can be calculated as a ratio of the measured $O_4$ dSCDs to the typical $O_4$ concentration at the altitude of the instrument $c_{O4}$ (instr):

$$dL_{eff}(O_4) = \frac{dSCD_{O_4}}{c_{O_4}(instr)} \tag{2}$$

As mentioned above, the profile shape of $O_4$ is an exponentially-decreasing profile with altitude. In contrast, the $NO_2$ profile

has a different shape, as this trace gas is emitted close to the surface. Therefore, the $dL_{eff}(O_4)$ cannot be used directly in Eq. (1) in order to estimate the $NO_2$ near-surface VMR. As indicated by Sinreich et al. (2013), Wang et al. (2014), and Ortega et al. (2015), the direct use of the $dL_{eff}$ derived from $O_4$ measurements introduces systematic errors in the near-surface $VMR_{NO2}$. In general, $dL_{eff}$ will be overestimated leading to an underestimation of the $VMR_{NO2}$ by up to a factor of three (Sinreich et al., 2013).

The introduction of a unit less correction factor ($f_c$) accounting for differences between the $O_4$ and $NO_2$ profile shapes is therefore necessary. $f_c$ connects the two different $dL_{eff}$ as follows:

$$dL_{eff}(NO_2) = dL_{eff}(O_4) \, f_c \tag{3}$$

Taking this relation into account and combining with Eq. (1), (2) and (3), the near-surface concentration of $NO_2$ can be expressed as:

$$c_{NO_2} = dSCD_{NO_2} \frac{c_{O_4 instr}}{dSCD_{O_4}} \frac{1}{f_c} \tag{4}$$

Previous studies (Ortega et al., 2015; Sinreich et al., 2013; Wang et al., 2014; Seyler et al., 2019) have highlighted the importance of properly estimating the correction factors, especially in polluted conditions, as those observed in Brussels. The $NO_2$ concentration corresponds to the ratio of $dSCD_{NO_2}$ to the light path length $dL_{eff}$ ($NO2$). Assuming that the $NO_2$ vertical distribution can be approximated by a box profile of height equal to MLH, one can also express it as the ratio between the

$VCD_{NO2}$ and the Mixing Layer Height (MLH):



$$\frac{dSCD_{NO_2}}{dL_{eff}(NO_2)} = \frac{VCD_{NO_2}}{MLH_{NO_2}} \tag{5}$$

Combining Eq. (3) and (5) and expressing the dSCD as the product of the VCD and a differential air mass factor (dAMF), it comes for $f_c$:

$$f_c = \frac{dAMF_{NO_2} MLH_{NO_2} c_{O_4}}{dAMF_{O_4} VCD_{O_4}} \tag{6}$$

where, $MLH_{NO_2}$ is the mixing layer height of $NO_2$, $dAMF_{NO_2}$, and $dAMF_{O4}$ are the $NO_2$ and $O_4$ differential air mass factor and $VCD_{O4}$ is the typical vertical column density of $O_4$ above the instrument. The dAMF of a trace gas expresses the light path enhancement with respect to the vertical path through the atmosphere. The correction factor depends on the aerosol load in the atmosphere, the solar zenith angle (SZA), the relative solar azimuth angle (RSAA), the MLH of the trace gas and the vertical distribution of the aerosols inside the MLH during the measurement.

To estimate MLH, we use the $NO_2$ vertical profile information derived in the main azimuthal direction. Assuming homogeneous mixing in the mixing layer, MLH is derived from the ratio of $VCD_{NO_2}$ to the near-surface concentration of $NO_2$. Moreover, during one MAX-DOAS scan, the vertical extent of the trace gas profile is considered homogeneous around the measurement site and the MLH values in the main azimuthal direction can also be applied to the other azimuthal directions. Despite its simplicity, this approach provides robust estimates of the MLH, consistent with local ceilometer observations (for

more details see Section 2.3.4). As indicated by Sinreich et al. (2013) and Ortega et al. (2015), the use of a realistic MLH daily variation is a crucial element in the parameterization method. Our approach represents an improvement over the more empirical approach used in previous studies.

The dAMF depends on the geometry (SZA, RSAA and elevation angle) as well as the aerosol and trace gas concentration profiles. For its calculation, we used VLIDORT (Spurr, 2006) version 2.7. The dAMF of $O_4$ and $NO_2$ were estimated for eight

different MLH scenarios (250 m -2000 m range) and for the Vis and UV wavelengths, separately. In these scenarios, the aerosol and $NO_2$ a priori profiles are specified as box profiles with a constant concentration from the surface to the MLH. The AOD vary from 0.30 to 0.60, the asymmetry parameter is set to 0.68 and the SSA to 0.92. The resulting correction factors are represented in the upper panels of Fig. 2 as a function of RSAA and for different values of the SZA (for an AOD equal to 0.30). They strongly depend on the RSAA and the MLH. For low RSAA and thick MLH $f_c$ reaches a maximum, while a

minimum is obtained at high RSAA for a thin MLH. When investigating the dependency of $f_c$ on the SZA for different AOD and RSAA values (lower panels in Fig. 2), we observe that it becomes highly dependent on AOD for low RSAA and SZA values close to 50-60°, indicating the limitations of the parameterization technique in those conditions.

The correction factor $f_c$ provides information about the state of the atmosphere, such that each measurement can be classified into one of the following three regimes. For $f_c$ equal or close to one, the effective light paths of $O_4$ and $NO_2$ are equal (Eq. 3),

which means that there is moderate to high aerosol load during the measurement. In contrast, when $f_c$ is significantly smaller



than one, the measurement is done under aerosol free conditions or thin MLH. Finally, $f_c$ can take values larger than one for cases of high SZA and low RSAA (Fig. 2), which are special conditions in which the parameterization method becomes highly dependent on the AOD. Such cases are highly uncertain and we exclude them from further analysis.

To estimate $f_c$ (Eq. 6) for every MAX-DOAS measurements in the Vis and UV wavelengths, $O_4$ and $NO_2$ dAMFs were
tabulated for eight different values of MLH (AOD set to 0.3) and for a suitable range of RSAA and SZA values. Using this look-up table, $O_4$ and $NO_2$ dAMFs are interpolated at the SZA, RSAA and MLH of each measurement. The near-surface VMR is then obtained by dividing the concentration of the trace gas (Eq. 4) by the air number density ($n_{air}$). For the calculation of $n_{air}$, the pressure and temperature profiles were taken from the AFGL 1976 Standard Atmosphere (Anderson et al., 1986) and are the same as used in Section 2.3.1. Furthermore, the VCD is estimated from the product of the near-surface concentration
with the MLH.

For the analysis, only measurements at SZA smaller than 80° were selected. As presented in Sinreich et al. (2013), the method is independent of the actual aerosol load, as long as a sufficient amount of aerosols is present in the troposphere (AOD>0.2). However, it depends slightly on the aerosol layer height. In order to select measurements where the near-surface layer can be parameterized as a box profile (i.e. with homogeneous concentration inside the layer), two conditions should be satisfied. First,
the scattering events corresponding to the lowest two elevation angles should occur in a comparable distance and secondly, those scattering events should happen inside the $NO_2$ layer, which can then be considered as homogeneous and therefore, parameterized as a box profile. In order to ensure that those conditions were satisfied, only scans for which the differences between $O_4$ and $NO_2$ dSCDs in the lowest two elevation angles were smaller than $10^{44}$ $molec^2$ $cm^{-5}$ and $10^{16}$ $molec$ $cm^{-2}$, respectively, were selected.

Furthermore, when the $O_4$ dSCD and, consequently the $dL_{eff}(O_4)$, is negative or too small because of bad weather conditions, the VMR can become unphysical (negative or close to zero). In consequence, measurements with a value of dLeff ($NO_2$) smaller

than 5 km (for both Vis and UV) are excluded from the study. An upper limit of 30 km is also adopted to exclude numerical outliers.

To estimate uncertainties on the retrieved $NO_2$ VMR and VCD using the parameterization method, two main error sources are considered: (1) uncertainties on $O_4$ and $NO_2$ dSCDs, and (2) uncertainties related to the estimation of the correction factors. Based on Eq. (4) and using a standard error propagation method, the overall uncertainty on the near-surface VMR is given by:

$$\sigma_{VMR}^2 = \left(\sigma_{dSCD}\frac{\partial VMR}{\partial dSCD}\right)^2 + \left(\sigma_{f_c}\frac{\partial VMR}{\partial f_c}\right)^2 \tag{7}$$

which results to:

$$\sigma_{VMR}^2 = \left(\sigma_{dSCD_{O_4}}\frac{VMR}{dSCD_{O_4}}\right)^2 + \left(\sigma_{dSCD_{NO_2}}\frac{VMR}{dSCD_{NO_2}}\right)^2 + \left(\sigma_{f_c}\frac{VMR}{f_c}\right)^2 \tag{8}$$

where:


$$\sigma_{f_c}^2 = \left(\sigma_{dAMF_{NO2}} \frac{f_c}{dAMF_{NO2}}\right)^2 + \left(\sigma_{dAMF_{O4}} \frac{f_c}{dAMF_{O4}}\right)^2 + \left(\sigma_{MLH} \frac{f_c}{MLH}\right)^2 \qquad (9)$$

Regarding the $\sigma_{dSCD}$, Bösch et al. (2018) and Kreher et al. (2019) indicated that, in urban or sub-urban polluted conditions, the DOAS fit uncertainty of both $O_4$ and $NO_2$ significantly underestimates the actual dSCD uncertainty, which is mostly driven by atmospheric variability, spatial and temporal fluctuations in the $O_4$ and $NO_2$ fields. In the present study, based on values

derived during the CINDI-2 campaign, conservative values of $3.5 \times 10^{15}$ molec cm$^{-2}$ ($NO_2$) and $1.5 \times 10^{42}$ molec2 cm$^{-5}$ ($O_4$) were used for the dSCD uncertainties in the UV and Vis ranges. This represents an error of up to 5 - 6 % on the $O_4$ dSCD and 4 – 5 % on the $NO_2$ dSCD in both the Vis and UV ranges.

The second important error source is related to the correction factors, which depend on air mass factor and $MLH_{NO2}$ calculations (Eq. 9). The uncertainty related to $MLH_{NO2}$ can be estimated as a combination of two terms: the total uncertainty of the $NO_2$

near-surface VMR and the $NO_2$ VCD derived by the MMF inversion algorithm. In the Vis range, the error related to $MLH_{NO2}$ is about 4 % and 5 % in the UV range. In order to estimate the uncertainty related to the air mass factor calculation, sensitivity tests about the input parameters in the RTM simulation were performed. In these sensitivity tests, the main inputs in the RTM calculations, such as the height of the assumed trace-gas profile and the aerosol properties, are modified. The corresponding dAMF variability is attributed to the uncertainty of the dAMF calculation. The error related to the $dAMF_{NO2}$ estimation is

about 2 % and 6 % in the UV and Vis ranges, respectively. The error related to $dAMF_{O4}$ is larger, reaching 18 % and 13 % in the Vis and UV ranges, respectively. Combining all error sources, the total uncertainties on the parameterized $NO_2$ are about 14 % and 20 % for the near-surface VMR and VCD in the visible range, while the corresponding errors in the UV are 7 % and 13 %. A summary of the above-mentioned error sources on the parameterized $NO_2$ is presented in Table 5.

### 2.3.3 Horizontal distribution of $NO_2$

A qualitative information about the horizontal distribution of $NO_2$ along each azimuthal direction can be obtained by considering how the dSCDs derived in the Vis and UV ranges depend on the retrieved horizontal light path lengths (dLeff (NO2)) (Ortega et al., 2015; Seyler et al., 2019). Indeed, dLeff (NO2) values depend strongly on scattering and atmospheric conditions and, since scattering processes are more pronounced at shorter wavelengths, dLeff (NO2) is shorter in the UV than in the Vis ($dL_{eff}(NO_2 \text{ Vis}) > dL_{eff}(NO_2 \text{ UV})$).

During one measurement, four useful pieces of information can be used in order to estimate the distance of the $NO_2$ concentration peak with respect to the instrument: the measured $NO_2$ near-surface VMR and the dLeff (NO2) in the Vis and UV ranges. Three different cases can be distinguished:

    (1)  $VMR_{NO2}(\text{Vis}) > VMR_{NO2}(\text{UV})$. In this case, the $NO_2$ peak ($dVMR_{NO2} = (dSCD_{Vis} - dSCD_{UV})/dL_{eff}(NO_2 \text{ UV})n_{air}$) is located further away from the measurement site and approximately, at the distance $dL_{eff}(NO_2 \text{ UV}) < dL < dL_{eff}(NO_2$

30           Vis).

    (2)  $VMR_{NO2}(\text{Vis}) < VMR_{NO2}(\text{UV})$. Here, the $NO_2$ peak ($dVMR_{NO2} = VMR_{NO2}(\text{UV})$) is located close to the MAX-DOAS instrument in a distance equal to $dL_{eff}(NO_2 \text{ UV})$.



(3) $VMR_{NO2}$ (Vis) = $VMR_{NO2}$ (UV). If both $NO_2$ VMR in the Vis and UV ranges are equal, it can be concluded that the $NO_2$ field ($dVMR_{NO2}$ = $VMR_{NO2}$ (Vis) = $VMR_{NO2}$ (UV)) is homogenously distributed along the line-of-sight.

This information is further exploited in Section 4.1, where the seasonal variation of the dual-scan MAX-DOAS measurements is presented.

### 2.3.4 Validation of the parameterization method

To validate the dual-scan parameterization method used in this study, two different approaches are adopted. First, the MLH, which is used in the calculation of the correction factors, is compared with MLH measurements using a co-located ceilometer. Second, the $NO_2$ near-surface VMRs and VCDs calculated by the parameterization technique in the main azimuthal angle (35.5° with respect to the North) is compared to corresponding results obtained with the MMF inversion algorithm.

To validate the MLH estimations, we use a co-located Vaisala CL51 ALC ceilometer operated by RMI. With this instrument, the MLH is retrieved according to an algorithm based on the direct analysis of backscatter gradient and variance (Haij at al., 2007; Haeffelin et al., 2016; Menut et al., 1999).

Figure 3 displays the diurnal variation of monthly-averaged MLH values derived from ceilometer and MAX-DOAS data during one full year, from March 2018 until March 2019. As can be seen, the MAX-DOAS data capture well the diurnal variation of the MLH measured by the ceilometer. The corresponding scatterplot is presented in Fig. 4. Both datasets are highly correlated (R=0.84), however the slope value (s=0.89) indicates that $MLH_{MAXDOAS}$ tend to slightly overestimate $MLH_{CEIL}$, the difference between both MLH values being generally smaller than 500 m. We note that the offset is larger during spring and summer. Since the ceilometer relies on the aerosol vertical distribution to derive MLH, and the MAX-DOAS uses the $NO_2$ vertical profile, differences in the absolute height values are expected. The $NO_2$ tropospheric columns, near-surface concentrations and atmospheric lifetime show a strong seasonality with maximum values during cold months. During warm months, the solar heating causes warmer air to rise. The typical time for air to rise from the surface to the top of the MLH is about 1 hr or less (Stull, 1988). During spring and summer, the differences between the ceilometer and MAX-DOAS MLH could be explained as follows: since the $NO_2$ lifetime is greater than 1 hr (Ehhalt et al., 1992) and in a combination with air uplifting activity (only present during warm seasons), $NO_2$ could be transported to higher heights than the ones estimated by the ceilometer.

The second approach to validate the parameterization technique consists of comparing the retrieved $NO_2$ near-surface VMR and VCD to the near-surface VMR (0 – 200 m) and VCD derived by using the MMF inversion algorithm. The only variable derived from MMF calculations and used in the parameterization technique is the MLH. As we can see in Fig. 5 and Fig. 6, results from both methods are highly correlated. The few cases where the two methods differ more substantially correspond to low $dAMF_{O4}$ values, which are associated to larger uncertainties.



## 3 Tropospheric NO₂ measurements from TROPOMI

Flying on board of the S5P satellite platform, the TROPOMI instrument is a passive grating imaging spectrometer covering the UV-Visible (270 - 500 nm), near-infrared (710 - 770 nm), and shortwave infrared (2314 - 2382 nm) spectral ranges (Veefkind et al., 2011). TROPOMI measures the solar back-scattered earthshine radiance in a push-broom configuration. With
a full swath width as wide as 2600 km, TROPOMI provides daily global coverage with a true-nadir pixel size of 7 x 3.5 km$^2$ in the UV/ Vis/ Near-Infrared bands. Since 06 August 2019, the TROPOMI spatial resolution is even higher with a pixel size of 5.5 x 3.5 km$^2$.

Developed at KNMI (Van Geffen et al., 2019), the tropospheric NO₂ algorithm uses a retrieval-assimilation-modelling system based on the 3-D global TM5 chemistry transport model. This retrieval scheme consists of three main steps. First, the total
NO₂ slant column density is retrieved from Level-1b radiance and irradiance spectra by applying the DOAS method. In a second step, the total NO₂ slant column density is separated into its stratospheric and tropospheric components by using the TM5-based data assimilation system. Finally, the tropospheric and stratospheric NO₂ slant column densities are converted to vertical column densities, by applying altitude-dependent AMFs. The AMF look-up tables are calculated on a 1º x 1º latitude-longitude grid using NO₂ vertical profiles from the TM5-MP model (Williams et al., 2017). They depend on the satellite
geometry, terrain height, cloud fraction, cloud height, and surface albedo.

The present study is based on RPRO and OFFL datasets of the TROPOMI L2 tropospheric NO₂ column product (see Table 6 for the corresponding versions). To focus on high quality measurements, only pixels characterized by a quality assurance value larger than 0.75 are used, so that pixels covered by clouds, snow or ice and erroneous retrievals are excluded from the analysis. The TROPOMI overpass over Brussels is around 13:30 LT. Figure 7 illustrates a typical comparison case. Superimposed on
TROPOMI ground-pixels as measured above Brussels on 06 June 2018, one can distinguish the different azimuthal viewing directions sampled by the Uccle dual-scan MAX-DOAS system. As can be seen, multiple pixels are coincident with one MAX-DOAS azimuthal direction, showing the high spatial resolution of TROPOMI as well as the challenges concerning its validation.

## 4 Results and discussion

### 4.1 Seasonal variation of dual-scan MAX-DOAS measurements

Box and whisker plots of MAX-DOAS horizontal effective light paths (dL$_{eff}$(NO₂), see Eq. 3) determined in each season for all the viewing azimuthal directions are presented in Fig. 8 for the Vis and UV wavelength ranges. As can be seen, the dL$_{eff}$ (NO₂) shows maximum median values during summer, and a minimum during winter, for both wavelength ranges. The seasonality of dL$_{eff}$(NO₂) is related to the seasonal variation of the aerosol content in the troposphere. For high aerosol load
conditions, the light path tends to become shorter due to increased scattering. In the Vis range, dL$_{eff}$(NO₂) can reach values of up to 19 km during winter with a mean value of 10 km, while in the UV, the maximum value is around 12 km with a mean



value of 8 km. Similar horizontal distance values have been found by Schreier et al. (2019b) using MAX-DOAS measurements in Vienna. Horizontal sensitivities ($dL_{eff}(NO_2)$) are generally larger in the Vis than in the UV, because of the more pronounced Rayleigh scattering at UV wavelengths.

In Fig. 9, the seasonal variation of the MAX-DOAS near-surface $NO_2$ VMR is presented for both Vis and UV channels in each azimuthal direction at 11:00 UTC. The length of the lines corresponds to the seasonally-averaged $dL_{eff}(NO_2)$. As mentioned above, $dL_{eff}(NO_2)$ is longer in the Vis than in the UV range, which leads to a more extended spatial sensitivity in the Vis than in UV. As explained in Section 2.3.3, the relationship between $NO_2$ VMRs and $dL_{eff}(NO_2)$ values contains information about the horizontal distribution of $NO_2$. This relationship as well as the wavelength dependence of the horizontal effective light path were also used by Seyler et al. (2019) to characterize the horizontal inhomogeneity of the $NO_2$ concentration above a shipping lane.

The near-surface $NO_2$ VMR has a clear seasonal cycle, with a maximum during winter due to higher emissions, lower temperature (and thus longer $NO_2$ lifetime) and shallower MLH, and a minimum in spring and summer. Moreover, the spatial distribution of $NO_2$ concentrations around Uccle shows a seasonal dependence. It should be noted that the main emission sources are located in the North and West part of the city and are associated mainly with the motorway around Brussels (the so-called Ring), the National Airport in Zaventem, and the Drogenbos power plant, the latter being located to the West of Uccle (Tack et al., 2017). In the absence of transport by the wind and given the fact that $NO_2$ has a shorter lifetime in the MLH (Beirle et al., 2011), the higher $NO_2$ concentrations should appear at the location of main emission sources. As can be seen in Fig. 9, this is not the case during all seasons, due to the influence of seasonal wind patterns. During winter, higher $NO_2$ concentrations are retrieved mainly in the North (N) and Northeast (NE) directions. The fact that the $NO_2$ concentration in the NE direction is higher in the Vis than in the UV, suggests that the $NO_2$ peak is located away from Uccle and closer to the Brussels National Airport. On the other hand, in the N direction, the $NO_2$ concentrations are lower in the Vis than in the UV. This can be associated with higher anthropogenic activity in the city center of Brussels. During spring and summer, the observed $NO_2$ VMRs are the lowest of the March 2018-March 2019 period. For spring, the maximum concentrations are measured away to the measurement site (Vis) in the South (S) and NW direction. It is worth mentioning that in the East (E) direction, the Vis and UV measurements have almost the same concentrations, indicating that the $NO_2$ field is homogeneous in those areas. During summer, the maximum $NO_2$ VMRs are retrieved in the Vis range and in the NE and NW directions, suggesting that the sources are mainly located away from the measurement site, possibly linked to the airport and city-center activity. One observes that in the S (in the direction of a large forested area), the retrieved concentrations are very low, while they are substantially higher in the NW and NE. Finally, during autumn, higher values are observed away from Uccle in the N direction, corresponding to sources located away from the measurement site, mostly traffic-related.

As already noted, the retrieved $NO_2$ VCDs can be influenced by the wind direction. In Fig. 10, MAX-DOAS $NO_2$ VCDs are represented as a function of the wind direction during the MAX-DOAS observations. The wind direction is measured by the BIRA-IASB meteorological station in Uccle. Generally, when the wind is blowing from the NE and SE direction, higher $NO_2$ columns are retrieved. During winter and autumn, the $NO_2$ VCDs, which are retrieved under different wind directions, differ



significantly compared to summer and spring. We conclude that emission sources located away from the measurement site influence the measured $NO_2$ concentration levels when wind is blowing in the direction of the site.

## 4.2 Comparison of MAX-DOAS and in-situ measurements

The in-situ telemetric air quality network (Bruxelles Environnement/Leefmilieu Brussel) of the Brussels region is used for

verifying the retrieved near-surface $NO_2$ VMR. Previous studies (e.g., Kramer et al., 2008; Schreier et al., 2019a)) have compared $NO_2$ MAX-DOAS measurements with in-situ concentrations, concluding to a considerable underestimation of $NO_2$ near-surface VMR by the MAX-DOAS instrument.

The present work uses hourly $NO_2$ near-surface concentrations from 10 monitoring stations distributed in the Brussels city area and provided by the Belgian Interregional Environment Agency (see http://www.irceline.be/en). Each station is

characterized according to its location: urban, urban background, traffic, rural or industrial. In Fig. 1, the location of the different in-situ stations is indicated together with the azimuthal viewing directions of the MAX-DOAS instrument.

In some directions, several in-situ stations are located at proximity of the MAX-DOAS line of sight, so that MAX-DOAS $NO_2$ concentrations can be compared to an average of the in-situ values reported at these stations. Because of the different spatial representativeness of the MAX-DOAS and in-situ techniques, one expects differences in the observed surface concentrations

(e.g., a VMR underestimation by the MAX-DOAS if the in-situ station is located close to a strong local emission source). However, for days where $NO_2$ is homogeneously distributed along the light path, both instruments should measure similar concentration levels. To restrict the comparison to conditions of good horizontal homogeneity, the hourly in-situ $NO_2$ near-surface concentrations for each in-situ station category were compared to each other and only measurements where the concentrations of the in-situ stations differed by less than 2 ppb within a time window of one hour were selected. Additionally,

we only considered MAX-DOAS measurements for which the horizontal sensitivity was less than 10 km. This distance is the maximum between the MAX-DOAS site and all the chosen in-situ stations.

Based on the different categories of in-situ stations, three groups were created similarly to the study of Kramer et al. (2008): (1) urban background, (2) urban and (3) traffic. The in-situ data were interpolated on the MAX-DOAS time grid and compared with the retrieved MAX-DOAS near-surface VMR. For the comparison, the in-situ dataset was averaged in bins of 2.5 ppb

length each. In Fig. 11, the results of this comparison show that the MAX-DOAS near-surface $NO_2$ VMRs have a systematic low bias when compared to in-situ data. As expected, the best agreement is found at urban background stations. For these sites, correlation coefficients of 0.81 and 0.95 and slope values of 0.35 and 0.40 are obtained for Vis and UV data, respectively. The majority of the comparison data points show that the MAX-DOAS $NO_2$ VMR are lower than the in-situ $NO_2$ observations by approximately a factor of two. However, a more careful inspection of the results indicates that a much better agreement is

obtained when considering $NO_2$ VMR values smaller than 12 ppb at urban background stations (R close to unity and slope close to 0.8). Such moderately polluted conditions likely correspond to homogeneously distributed $NO_2$ fields similarly sampled by in-situ and remote-sensing measurements.



The worst agreement is found at traffic and urban stations (R in the 0.53-0.62 range). At these stations, the MAX-DOAS $NO_2$ VMRs are lower than the in-situ $NO_2$ observations by approximately a factor of two and three, respectively. Two reasons can explain these findings. First, MAX-DOAS concentrations are integrated along a long light path, which smooths out the variability of the $NO_2$ field along this path, while the in-situ instruments perform measurements at a single location point. Secondly, the in-situ stations are located typically at 3-10 m altitude, while the MAX-DOAS near-surface measurements correspond to a layer from the surface to 200 m altitude (according to the selected altitude grid in the aerosol and $NO_2$ OEM-based profile retrievals) and hence, is not fully representative of the surface concentration. Similar findings were reported in Kramer et al. (2008).

### 4.3 Validation of TROPOMI tropospheric $NO_2$ columns

To validate the TROPOMI tropospheric $NO_2$ columns, we adopted three successive approaches:

1) A first comparison is performed by selecting only MAX-DOAS data in the main azimuthal direction (35.5° w.r.t N) and the TROPOMI value from the closest pixel located in the same direction as the MAX-DOAS measurement.

2) To improve the spatial coincidence between MAX-DOAS and TROPOMI observations, a second comparison is performed by using the dual-scan MAX-DOAS observations: measurements in every MAX-DOAS azimuthal direction are compared with a weighted average of TROPOMI columns as measured in coincident pixels with the weighting being determined by the MAX-DOAS horizontal segment crossing every pixel.

3) The impact of possible systematic uncertainties in the satellite retrieval (in particular the a-priori profile shape) is investigated.

In order to increase the number of co-location pairs, we compare both UV and Vis MAX-DOAS measurements with TROPOMI. It should be emphasized that UV and Vis MAX-DOAS VCDs correspond to different dLeff (NO2) values (see Fig. 8 and Section 2.3.3) and consequently, different sampling of TROPOMI pixels. In the following sections, the regression analysis parameters (correlation coefficient and slope value) refer to both UV and Vis MAX-DOAS measurements together.

### 4.3.1 Validation based on one MAX-DOAS azimuthal direction measurements

In this first approach, the MAX-DOAS tropospheric $NO_2$ VCDs, derived in the main azimuthal direction by applying the MMF algorithm, are compared to TROPOMI observations in the same direction as the MAX-DOAS measurement. We select the closest TROPOMI pixel that intercept the MAX-DOAS line-of-sight within a radius of 20 km around Uccle. This approach has generally been used in previous satellite validation studies based on MAX-DOAS observations (e.g., Chen et al., 2009; Irie et al., 2008; Ma et al., 2013). It is adopted for reference against other comparison strategies that make use of more than one azimuthal MAX-DOAS measurement (see section 4.3.2).

We compare TROPOMI daily measurements with MAX-DOAS $NO_2$ VCDs averaged around ($\pm$ 1 hr) of the TROPOMI overpass time. A first comparison plot is presented in Fig. 12, where the time series of the TROPOMI tropospheric $NO_2$ VCD is displayed together with the Vis MAX-DOAS measurements. All TROPOMI and MAX-DOAS data points presented in Fig.





12 satisfy the quality check requirements for both datasets (see Section 2.3.1 and 3). The MAX-DOAS error bars represent the standard deviation of the mean values ($\pm$ 1 hr), while the TROPOMI bars are equal to the tropospheric column error of the pixel. Results show that MAX-DOAS measurements have a larger variability than corresponding TROPOMI observations. In addition, TROPOMI tropospheric $NO_2$ columns are systematically lower than co-located MAX-DOAS data.

Figure 13 presents scatter plots of TROPOMI tropospheric $NO_2$ VCDs against MAX-DOAS data for the four seasons from March 2018 to March 2019. The highest correlation is found during spring (R=0.71) and summer (R=0.64) seasons, while lower correlation are obtained in winter and fall, with correlation coefficient values in the 0.28-0.63 range. During autumn 2018, the number of Vis data points is significantly smaller than in the UV because of more frequent unrealistic retrieved profiles. At the other hand, during summer, the accepted scans are almost equal for the UV and Vis ranges. The same trend of

accepted scans to the total number of scans is observed during autumn and summer 2019, concluding that the meteorological conditions (mostly cloud cover) during autumn affect strongly the Vis retrieval. Regarding the slope values, they are all smaller than 0.5, except for spring, indicating that TROPOMI columns are about a factor of two lower than MAX-DOAS columns (in agreement with the S5P MPC VDAF Validation Web Article, available at: http://mpc-vdaf.tropomi.eu/).

These results indicate that the discrepancy between TROPOMI and MAX-DOAS measurements is significant during all

seasons and particularly marked during winter and autumn. A possible explanation could be due to differences in the air masses probed by the two instruments. The use of only one satellite pixel, even if its direction with respect to the MAX-DOAS line of sight is taken into account, is not necessarily the most appropriate comparison method. One azimuthal MAX-DOAS measurement samples air masses along a light path of several kilometers in a fixed direction, which corresponds to more than one TROPOMI pixel, as outlined in Fig. 7. One expects this horizontal sampling effect to be more marked in winter and fall,

given the larger $NO_2$ concentration gradients observed during these seasons compared to the other ones (see Section 4.1 and Fig. 9).

### 4.3.2    Validation based on dual-scan MAX-DOAS measurements

In a second step, we compare TROPOMI tropospheric $NO_2$ columns with the dual-scan parameterized MAX-DOAS $NO_2$ VCDs. Two improvements are introduced: (1) the use of more than one MAX-DOAS azimuthal direction and (2) a better

spatial selection of the TROPOMI pixels accounting for the MAX-DOAS horizontal sensitivity ($dL_{eff}(NO_2)$). Only satellite pixels located along the segments of length $dL_{eff}(NO_2)$ in the different MAX-DOAS azimuthal directions and timely coincident dual-scan MAX-DOAS observations (TROPOMI overpass time $\pm$ 1 hr) are selected. MAX-DOAS $NO_2$ VCDs in every MAX-DOAS azimuthal direction are compared to a weighted average of TROPOMI columns from the different pixels that are crossed by the corresponding MAX-DOAS horizontal line of sight segment. The weight of a given pixel is derived from the

length of the segment portion that crosses the pixel. We can use Fig. 7 as a simplified scheme to show two important aspects of this comparison: (1) every MAX-DOAS azimuthal line-of-sight is representative for a segment section that extends from the instrument to a distance equal to $dL_{eff}(NO_2)$ and (2), $dL_{eff}(NO_2)$ in one azimuthal direction can be separated into different portions that cross each of the selected satellite pixels.





Results displayed in Fig. 14 show that the agreement between TROPOMI and MAX-DOAS datasets is significantly improved, especially in terms of correlation (R in the 0.65-0.82 range). Owing to the improved spatial coincidence associated to the use of dual-scan MAX-DOAS data and the better spatial coincident criterion between TROPOMI and MAX-DOAS data, the scatter in the data points is also substantially reduced during all seasons and especially in winter. Another interesting feature

is the improvement of the slope values, observed in all seasons (slopes in the 0.41-0.71 range). During seasons with a more homogeneous $NO_2$ field, the improvement of the slope values is less pronounced than during seasons, like winter and autumn, where the $NO_2$ field can be highly inhomogeneous. During spring, the slope value is reduced despite the better correlation. Overall, TROPOMI still underestimates MAX-DOAS measurements by about 40-50 %.

## 4.4 Investigation of systematic uncertainties in TROPOMI $NO_2$ retrievals

To identify the origin of the persisting underestimation of TROPOMI $NO_2$ measurements, we investigated the most relevant sources of uncertainties in the satellite retrievals.

Boersma et al. (2004) presented a thorough analysis of satellite tropospheric $NO_2$ column retrieval uncertainties. Main error sources are related to the spectral fitting (dominated by measurement noise), the estimation of the stratospheric $NO_2$ column and knowledge of the main ancillary parameters used for the AMF calculation, i.e. surface albedo, cloud fraction and cloud

top height, aerosols, and the a-priori $NO_2$ profile shape. In the following sub-sections, we briefly discuss uncertainties related to cloud, aerosol and surface albedo and afterwards, we investigate in more details the role of the a-priori $NO_2$ profile.

### 4.4.1 Clouds and aerosols

Clouds can have a major impact on tropospheric $NO_2$ observations from space, because of their strong influence on the incoming solar radiation (Boersma et al., 2004; Koelemeijer et al., 2001). In the TROPOMI tropospheric $NO_2$ retrieval

algorithm, only cloud-free and weakly cloudy scenes are considered as valid measurements satisfying the recommended quality assurance value (QA>= 0.75) (see Section 3). Although this quality flagging effectively minimizes uncertainties due to clouds on the $NO_2$ product, many selected scenes are still partially cloud-covered and affected by cloud-related errors. In the TROPOMI processor, clouds are characterized by using cloud fraction and cloud top height parameters, which are both derived from radiance observations in the $O_2$ A-band. This cloud information is used as an input in a cloud-correction scheme applied

to $NO_2$ retrieval (van Geffen et al., 2019). Cloud-induced errors are complex and can lead to positive or negative biases on the tropospheric $NO_2$ column. As a result, cloud-induced errors are generally pseudo-random in nature. So except for specific cases (e.g. persisting contamination by heavy aerosol or cloud layers), cloud-related errors can hardly account for systematic biases in $NO_2$ retrievals.

Like clouds, aerosols can affect the accuracy of tropospheric $NO_2$ retrieval from space (Heckel et al., 2011; Leitão et al., 2010;

McLinden et al., 2014). In the TROPOMI $NO_2$ algorithm, aerosols are not explicitly treated which means that all AMF calculations are performed for a Rayleigh atmosphere (clouds being treated as simple Lambertian reflectors). The impact of aerosols is however considered indirectly through the cloud correction algorithm, under the assumption that scattering aerosols





will tend to increase the cloud fraction. For non-absorbing aerosols of moderate optical thickness, like typically observed in Brussels, this simplified approach was shown to be effective in accounting for the impact of reflecting aerosols on tropospheric $NO_2$ AMFs (Boersma et al., 2011).

### 4.4.2 Surface Albedo

Surface albedo is another parameter having a significant influence on satellite tropospheric $NO_2$ AMFs. In the study of Boersma et al. (2004), it was shown that the $NO_2$ AMF sensitivity to albedo is large especially for albedos smaller than 0.2. For albedo values between 0.0 and 0.2, which are common in the blue spectral range over land, a difference of 0.015 in the surface albedo can lead to a 12 % change of the tropospheric $NO_2$ AMF. In order to estimate surface albedo uncertainties in the Uccle conditions, observation from the Airborne Prism Experiment (APEX) performed above Brussels during June 2015

(Tack et al., 2017) were compared with climatological values used in the TROPOMI operational algorithm. The difference between these two independent estimates of the albedo were found to be small in average (smaller than 0.01) suggesting that albedo data used in the TROPOMI algorithm are well representative of the Brussels area in June.

However, we should keep in mind that the surface albedo values used in the TROPOMI retrieval have a spatial resolution of 13 x 24 km$^2$ (OMI spatial resolution). In reality, inside an area of 13 x 24 km$^2$ in an urban environment, we expect to have

considerable differences between the albedo values at the scale of TROPOMI pixels. So even if the difference between APEX and TROPOMI albedos was found to be small in average for the June 2015 flight, further investigation is needed to fully assess the impact of albedo uncertainties on the TROPOMI $NO_2$ product (F. Tack, personal communication, 2020).

### 4.4.3 A-priori $NO_2$ profile shape

The TROPOMI $NO_2$ retrieval algorithm $NO_2$ vertical profiles specified by the TM5-MP model, for 34 vertical layers at the

horizontal resolution of 1$^o$ x 1$^o$ in latitude-longitude (Williams et al., 2017). In comparison to the TROPOMI pixel size (3.5 x 7 km$^2$), the resolution of TM5-MP (approximately 100 x 100 km$^2$) is thus very coarse and cannot capture spatial gradients at the scale of a city like Brussels.

A way to test how uncertainties on the a-priori profile influence the TROPOMI $NO_2$ VCDs in our observation conditions is to use vertical profiles derived from our MAX-DOAS measurements to recalculate the satellite $NO_2$ VCDs. In order to perform

this transformation, we use the Averaging Kernels (AK) information provided in the TROPOMI $NO_2$ product. The AK describes how the sensitivity of the retrieval depends on altitude. For satellite measurements of tropospheric species in the UV-Vis range, the AK generally increases with altitude in the first kilometers above the surface (Fig. 15). Since the $NO_2$ profile has its maximum close to the surface, accurate knowledge of the $NO_2$ vertical distribution in this altitude range is therefore critical for the calculation of the $NO_2$ AMFs.

Using the formula described in Appendix A and daily-averaged MAX-DOAS concentration profiles derived in the main azimuthal direction using the MMF algorithm, a modified version of the TROPOMI tropospheric $NO_2$ column product was generated. Daily-averaged MAX-DOAS profiles were used to minimize the impact of instabilities frequently observed in individual profile retrievals, as illustrated in Fig. 16.



Figure 17 presents validation results corresponding to the recalculated TROPOMI $NO_2$ columns. Comparing with results from Fig. 14, one can see that the change in $NO_2$ profile shape has a strong impact on validation results, leading to a better agreement between satellite and ground-based data sets. During all seasons, the slopes of the linear regressions are largely improved (slopes in the 0.81-1.16 range), which essentially resolves the previously reported underestimation. In average, the recalculated

TROPOMI columns increase by about 55 %. Looking more closely at Figs. 14 and 18, one can see that the application of MAX-DOAS $NO_2$ vertical profiles mostly improves the agreement for tropospheric $NO_2$ columns larger than 1.0 $10^{16}$ molec $cm^{-2}$ (for the cases with $NO_2$ enhancement). One can also note that correlation coefficients are slightly degraded after application of the MAX-DOAS profiles, suggesting that the applied transformation introduces some more scatter in the comparison. Table 7 presents a detailed summary of all the regression analyses conducted.

In conclusion, the change of the a-priori profile in the TROPOMI retrieval has a significant impact on the agreement between the satellite and MAX-DOAS measurements, leading to a satisfying closure of the validation study. Although based on a different approach, these results are in agreement with the recent studies of Ialongo et al. (2019) and Judd et al. (2019).

## 5 Conclusions

One year of S5P/ TROPOMI tropospheric $NO_2$ columns recorded above Brussels were validated using dual-scan MAX-DOAS

measurements. The MAX-DOAS instrument was installed in Uccle, a sub-urban site, located in the south of Brussels-Capital Region. A standard acquisition scheme was implemented combining vertical scans in a fixed azimuthal direction (main azimuthal direction pointing to Brussels Airport) and horizontal scans in ten azimuth angles at a fixed elevation angle (2°). OEM-based profile retrievals were performed in the main azimuthal direction and a parameterization technique, based on Sinreich et al. (2013) was applied in all the other azimuthal directions to retrieve dual-scan $NO_2$ near-surface VMRs and VCDs.

An appropriate characterization of the MLH was obtained by using the vertical profile inversion results in the main azimuthal direction.

The dual-scan parameterized $NO_2$ VMRs and VCDs were validated using ancillary measurements. Three different comparisons were carried out: (1) the MAX-DOAS-based MLH values used in the parameterization were compared with measurements from a co-located ceilometer instrument, (2) the parameterized $NO_2$ near-surface VMRs and VCDs retrieved in the main

azimuthal direction were compared with the same quantities derived from OEM-based profiles, and (3) the dual-scan $NO_2$ near-surface VMRs were compared with in-situ $NO_2$ concentrations. A good overall agreement was found for both comparisons (UV and Vis datasets) during the whole year of measurements.

The seasonal variability of the $NO_2$ VMR around the measurement site was investigated. As expected, higher $NO_2$ concentrations are observed during winter due to larger emissions, a shallower MLH, and lower temperatures resulting in

longer lifetimes. Wind speed and direction are also found to play a significant role on the distribution of $NO_2$ around the site. As the main emission sources are located to the north of Uccle, concentration peaks are associated with wind blowing mainly from the NE direction. The dual-scan MAX-DOAS retrievals were also compared to $NO_2$ measurements from the in-situ air





quality telemetric network of the Brussels region. For this comparison, in-situ stations were selected along the different MAX-DOAS azimuthal directions. Although the in-situ measurements show systematically larger values than those derived from the MAX-DOAS instrument, a good correlation is found between both data sets, especially for urban background sites under moderately polluted conditions.

In a second step, MAX-DOAS data were used to validate TROPOMI tropospheric $NO_2$ measurements. Two different approaches were used. First, the MAX-DOAS $NO_2$ VCDs, derived by applying an OEM-based inversion algorithm in the main azimuthal direction, were compared with the closest TROPOMI pixel, located along the main MAX-DOAS pointing direction. Results show a clear seasonal behavior and a tendency for satellite data to underestimate the MAX-DOAS tropospheric $NO_2$ columns during all seasons. In the second approach, the dual-scan parameterized MAX-DOAS tropospheric $NO_2$ columns and

corresponding effective horizontal distances were used to define a wider sampling area around the measurement site for the selection of the TROPOMI pixels. MAX-DOAS measurements in every azimuthal direction were compared to a weighted average of TROPOMI columns appropriately selected for optimal matching with MAX-DOAS observation directions and effective horizontal distances. Although dual-scan MAX-DOAS measurements lead to improved agreement with satellite data, a systematic underestimation of the TROPOMI tropospheric columns is still observed.

Further, a detailed investigation of the main ancillary parameters used for the AMF calculation in the TROPOMI tropospheric $NO_2$ columns retrievals revealed that the a-priori $NO_2$ profile shape uncertainty has a large impact on the satellite measurements. Recalculating the TROPOMI columns using daily median MAX-DOAS profiles as a priori results in a much better agreement between satellite and MAX-DOAS data. This suggests that the use of more appropriate a priori profiles in the TROPOMI retrieval can improve substantially the accuracy of the satellite tropospheric $NO_2$ data, especially in urban areas.

The improvement is however less clear during seasons characterized by highly variable $NO_2$ fields and cloudy conditions.

In conclusion, our study shows that dual-scan MAX-DOAS measurements conducted in an urban area offer (1) the possibility to better characterize the spatial variability of short-lived pollutants like $NO_2$, and (2) to improve the validation of satellite measurements in an urban environment. Moreover, the vertical profiling capability of MAX-DOAS measurements allows testing the suitability of the a priori profile shape information used in satellite retrievals. Based on our results, additional work

could be done for improving future TROPOMI validation exercises. For instance, the horizontal resolution of the satellite a-priori profiles could be further improved by performing vertical MAX-DOAS scans during TROPOMI overpass in more than one azimuthal direction. Additionally, the satellite retrieval uncertainties related to clouds and aerosols could be also investigated into more details based on the azimuthal scan capability of MAX-DOAS instruments.

**Appendix A: Profile-shape adjustment of TROPOMI $NO_2$ VCDs**

We start from the general formula used to derive the $NO_2$ VCD from satellite measurements (e.g. van Geffen et al., 2014):

$$VCD_{SAT} = \frac{SCD_{SAT}}{AMF_{SAT}} \tag{A1}$$





Where $SCD_{SAT}$ stands for the $NO_2$ slant column density, and $AMF_{SAT}$ for the $NO_2$ air mass factor as used in the operational algorithm, i.e. based on a-priori $NO_2$ vertical profiles specified by the TM5 chemistry-transport model.

For optically thin conditions valid in the blue spectral range where $NO_2$ is retrieved, the satellite AMF ($AMF_{SAT}$) can be expressed as a linear sum of layer (or box) air mass factors ($AMF_i^{SAT}$), weighted by the $NO_2$ VCD contribution in each atmospheric layer:

$$AMF_{SAT} = \frac{1}{VCD_{a-priori}} \sum_i AMF_i^{SAT} C_i^{a-priori} \tag{A2}$$

where $C_i^{a-priori}$ represents the a-priori $NO_2$ partial column in atmospheric layer $i$.

In addition, the vertical sensitivity of the $NO_2$ retrieval is given by the averaging kernel (AK) according to:

$$AK_i^{SAT} = \frac{AMF_i^{SAT}}{AMF_{SAT}} \tag{A3}$$

When comparing satellite and ground-based measurements (here from a MAX-DOAS instrument), it is a good practice to smooth the ground-based reference profile using the satellite AK (see e.g. Eskes and Boersma, 2003):

$$VCD_{MAXDOAS}^{smoothed} = \sum_i AK_i^{SAT} c_i^{MAXDOAS} \tag{A4}$$

An alternative approach is to recalculate the satellite VCD using the MAX-DOAS profile as a-priori in the satellite retrieval. Only the AMF is concerned and, similarly to equation (A2), we can write:

$$AMF_{SAT}^{MAXDOAS-pro} = \frac{1}{VCD_{MAXDOAS}} \sum_i AMF_i^{SAT} C_i^{MAXDOAS} \tag{A5}$$

or, using equations (A3) and (A4):

$$AMF_{SAT}^{MAXDOAS-pro} = \frac{AMF_{SAT}}{VCD_{MAXDOAS}} \sum_i AK_i^{SAT} C_i^{MAXDOAS} = AMF_{SAT} \frac{VCD_{MADOAS}^{smoothed}}{VCD_{MAXDOAS}} \tag{A6}$$

which finally leads to:

$$VCD_{SAT}^{MAXDOAS-pro} = VCD_{SAT} \frac{VCD_{MAXDOAS}}{VCD_{MAXDOAS}^{smoothed}} \tag{A7}$$

*Data Availability.* The datasets generated and analyzed in the present work are available from the corresponding author on request.

*Author contributions.* ED undertook the development and validation of the dual-scan MAX-DOAS retrieval strategy in Uccle, exploited the MAX-DOAS retrievals during one year, performed the validation of TROPOMI tropospheric $NO_2$ columns and wrote the manuscript. FH supported and guided ED in the MAX-DOAS retrieval exploitation as well as in the different TROPOMI validation approaches and revised and edited the manuscript. GP provided the dataset of the TROPOMI tropospheric $NO_2$ columns and supported ED in the TROPOMI validation approaches. MMF provided the MMF inversion algorithm and the RTM, supported and guided ED in the aerosol, $NO_2$ OEM-based profile retrievals and dAMF forward



calculations. AM and FT contributed in scientific discussions and the manuscript revision. HDL tested different TROPOMI validation approaches in Uccle. CF and CH provided technical and software support for the MAX-DOAS instrument in Uccle. CF developed the QDOAS software and guided ED in the DOAS analysis. QL provided the MLH dataset derived by the ceilometer. FF provided useful information about the in-situ dataset. MVR supervised the present work, provided general

guidelines and valuable comments during the whole process of the manuscript preparation, revised and edited the manuscript. All authors reviewed, discussed the results and commented on the manuscript.

*Competing interests.* The authors declare that they have no conflict of interest.

*Acknowledgements.* We gratefully acknowledge the Belgian Federal Science Policy Office (BELSPO) for funding this study (Supplementary Researcher grant).

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



**Table 1: MAX-DOAS experimental set-up.**

| Date | Scan-mode | Azimuth angle (º with respect to the North/Eastward direction) | Elevation angle (º) | Scan duration (min) |
|---|---|---|---|---|
| 01 Mar. 18 – 20 Jun. 18 | Vertical | 35.5 | 0, 1, 2, 3, 4, 5, 6, 8, 12, 30, 90 | 11 |
| | Horizontal | 330.5, 302.5, 227.5, 167.5, 86.5 | 0, 1, 2 | 15 |
| 20 Jun. 18 – 19 Oct. 18 | Vertical | 35.5 | 0, 1, 2, 3, 4, 5, 6, 8, 12, 30, 90 | 11 |
| | Horizontal | 25, 32, 348, 305, 302.5, 300, 265, 262.5, 180, 167.5, 123.5, 105, 75, 62.5 | 2 | 14 |
| 19 Oct. 18 - ongoing | Vertical | 35.5 | 1, 2, 3, 4, 5, 6, 8, 12, 30, 90 | 10 |
| | Horizontal | 11, 25, 32, 62.5, 105, 262.5, 305, 353, 344 | 2 | 9 |

**Table 2: DOAS settings for $NO_2$ and $O_4$ in the Vis spectral range.**

| Wavelength range | 425-490 nm |
|---|---|
| Fraunhofer reference spectra | Noon zenith spectra |
| **Cross-sections:** | |
| $NO_2$ (294 K) | Vandaele et al. (1998) with $I_0$ correction (SCD of $10^{17}$ molecules/cm²) |
| $NO_2$ (220 K) | Pre-orthogonalized Vandaele et al. (1998) with $I_0$ correction (SCD of $10^{17}$ molecules/cm²) |
| $O_3$ (223 K) | Serdyuchenko et al. (2014) with $I_0$ correction (SCD of $10^{20}$ molecules/cm²) |
| $O_4$ (293 K) | Thalman and Volkamer (2013) |
| $H_2O$ | HITEMP (Rothman et al., 2010) |
| Ring | RING_QDOAS_SAO2010 |





**Table 3: Same as Table 2 for the UV spectral range.**

| Wavelength range | 338-370 nm |
|---|---|
| **Fraunhofer reference spectra** | Noon zenith spectra |
| **Cross-sections:** | |
| NO$_2$ (298 K) | Vandaele et al. (1998) with I$_0$ correction (SCD of $10^{17}$ molecules/cm$^2$) |
| NO$_2$ (220 K) | Pre-orthogonalized Vandaele et al. (1998) with I$_0$ correction (SCD of $10^{17}$ molecules/cm$^2$) |
| O$_3$ (223 K) | Serdyuchenko et al. (2014) with I$_0$ correction (SCD of $10^{20}$ molecules/cm$^2$) |
| O$_3$ (243 K) | Pre-orthogonalized Serdyuchenko et al. (2014) with I$_0$ correction (SCD of $10^{20}$ molecules/cm$^2$) |
| O$_4$ (293 K) | Thalman and Volkamer (2013) |
| HCHO (297 K) | Meller and Moortgat (2000) |
| BrO (223 K) | Fleischmann et al. (2004) |
| Ring | RING_QDOAS_SAO2010 |

**Table 4: Error budget overview of the MMF retrieved NO$_2$ VMR and VCD in the Vis and UV spectral ranges. The total uncertainty is calculated as the square root of the sum of the squares of the different error sources.**

| Error overview (%) | NO$_2$ VMR (VIS) | NO$_2$ VCD (VIS) | NO$_2$ VMR (UV) | NO$_2$ VCD (UV) |
|---|---|---|---|---|
| **Noise error** | 2 | 2 | 2 | 3 |
| **Smoothing error** | 3 | 8 | 2 | 9 |
| **Forward model uncertainty** | 3 | 4 | 2 | 1 |
| **Uncertainty on NO$_2$ cross-sections** | 3 | 3 | 3 | 3 |
| **Uncertainty related to the temperature dependence of NO$_2$ cross-sections** | 9 | 9 | 9 | 9 |
| **Total uncertainty** | 11 | 13 | 10 | 14 |





**Table 5: Error budget overview of the parameterized NO$_2$ VMR in the Vis and UV spectral ranges.**

| Error overview (%) | VIS | UV |
|---|---|---|
| NO$_2$ DOAS fit | 4 | 5 |
| O$_4$ DOAS fit | 5 | 6 |
| MLH | 4 | 5 |
| dAMF$_{NO2}$ | 2 | 6 |
| dAMF$_{O4}$ | 18 | 13 |
| Total uncertainty on the VMR | 14 | 7 |
| Total uncertainty on the VCD | 20 | 13 |

**Table 6: TROPOMI NO$_2$ processor versions used in this study.**

| Dataset | Number of version | Starting date of operation | End date of operation |
|---|---|---|---|
| RPRO | 010202 | 17/03/2018 | 17/10/2018 |
| OFFL | 010200 | 17/10/2018 | 27/11/2018 |
| OFFL | 010202 | 28/11/2018 | 20/03/2019 |



**Table 7: Summary table of the regression analysis parameters derived by the three validation exercises.**

| TROPOMI dataset | MAX-DOAS dataset (Vis/UV together) | Season | Correlation coefficient (R) | Slope (s) |
|---|---|---|---|---|
| - One pixel<br>- Closest pixel to the measurement site<br>- In the same direction as the MAX-DOAS line-of-sight | One azimuthal direction | Winter<br>Spring<br>Summer<br>Spring | 0.28<br>0.71<br>0.64<br>0.63 | 0.34<br>0.97<br>0.55<br>0.31 |
| - More than one pixel<br>- Weighted average of satellite columns<br>- In the same direction as the different MAX-DOAS line-of-sights | Multiple MAX-DOAS azimuthal directions | Winter<br>Spring<br>Summer<br>Spring | 0.65<br>0.80<br>0.80<br>0.82 | 0.71<br>0.41<br>0.53<br>0.58 |
| - More than one pixel<br>- Use of MAX-DOAS a-priori profiles to recalculate the satellite columns<br>- Weighted average of recalculated satellite columns<br>- In the same direction as the different MAX-DOAS line-of-sights | Multiple MAX-DOAS azimuthal directions | Winter<br>Spring<br>Summer<br>Spring | 0.58<br>0.67<br>0.74<br>0.70 | 1.13<br>0.81<br>1.16<br>0.86 |



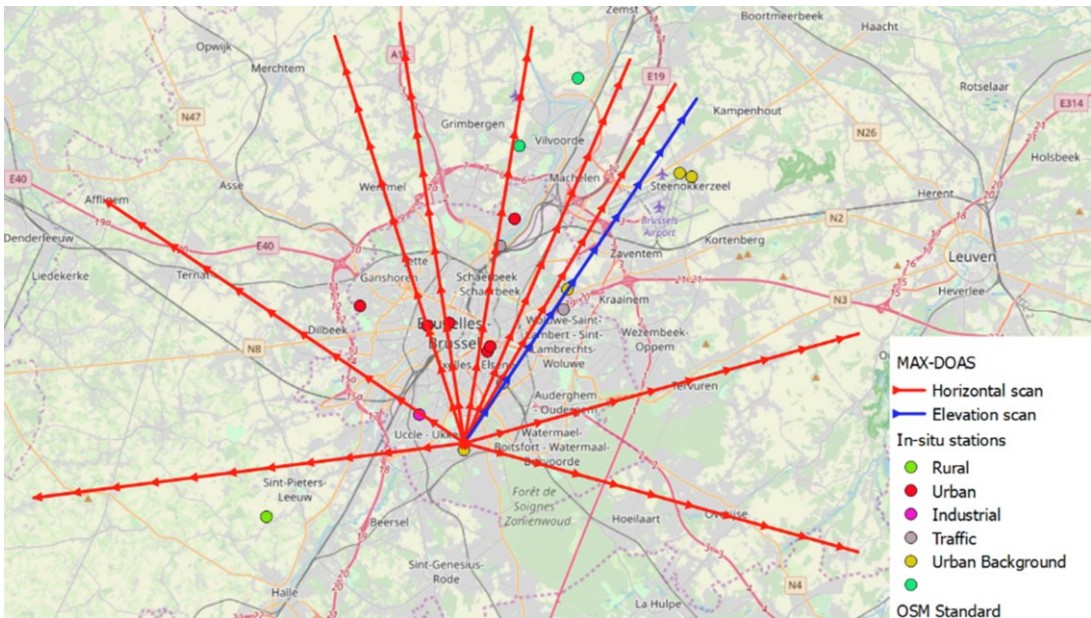

**Figure 1: The final dual-scan experimental set-up of the BIRA-IASB MAX-DOAS instrument (overlaid onto Open Street Map (OSM) Standard layer). The blue line represents the vertical scan mode and the red lines represents the horizontal scan mode, with a line length of 20 km each. The colored dots show the different types of in-situ stations around the MAX-DOAS instrument (see Section 4.2). © OpenStreetMap contributors 2019. Distributed under a Creative Commons BY-SA License.**



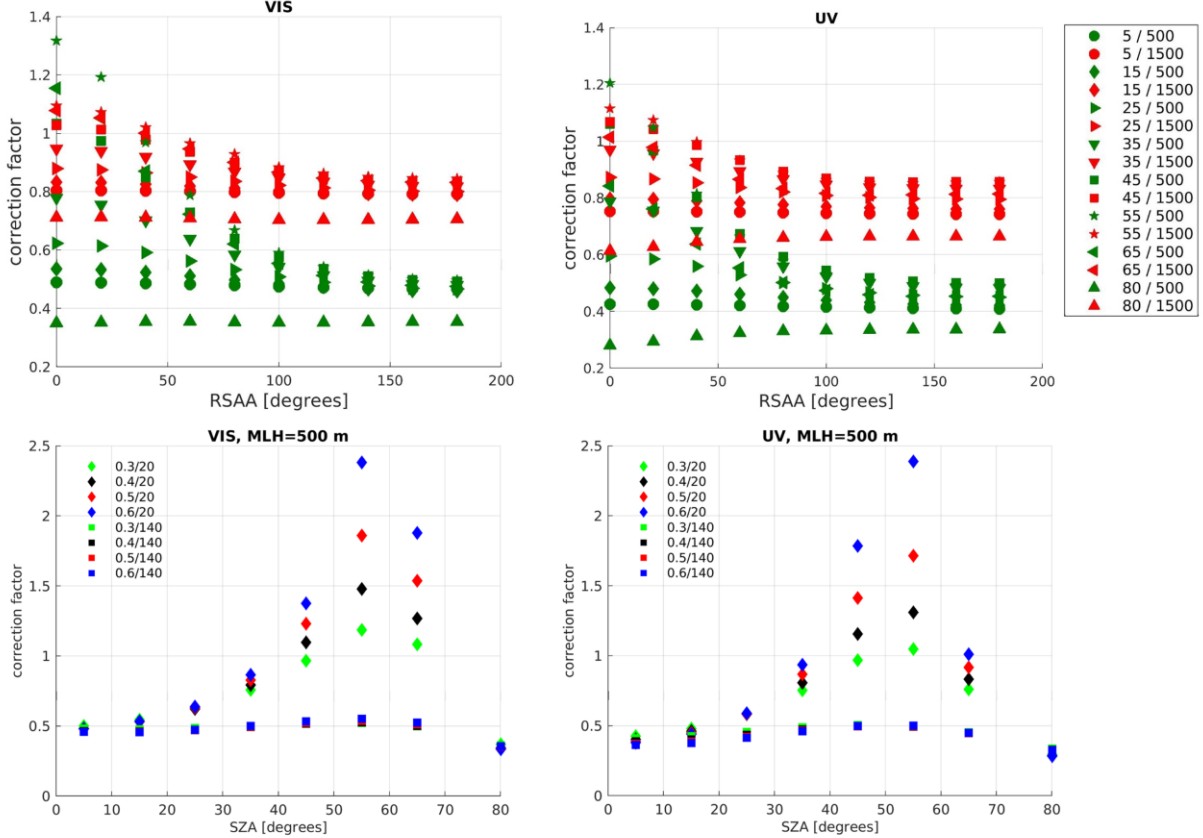

**Figure 2: Upper panels correspond to the correction factors as a function of RSAA for two different MLH values (in every RSAA and MLH values, the different data points correspond to different SZA values) in the Vis and UV wavelength ranges. The first value in the symbols list corresponds to the SZA and the second to the MLH. Lower panels show the correction factors as a function of SZA for different AOD scenarios and RSAA values in the Vis and UV ranges for a MLH set to 500 m. The first value in the symbols list corresponds to the AOD and the second to the RSAA.**



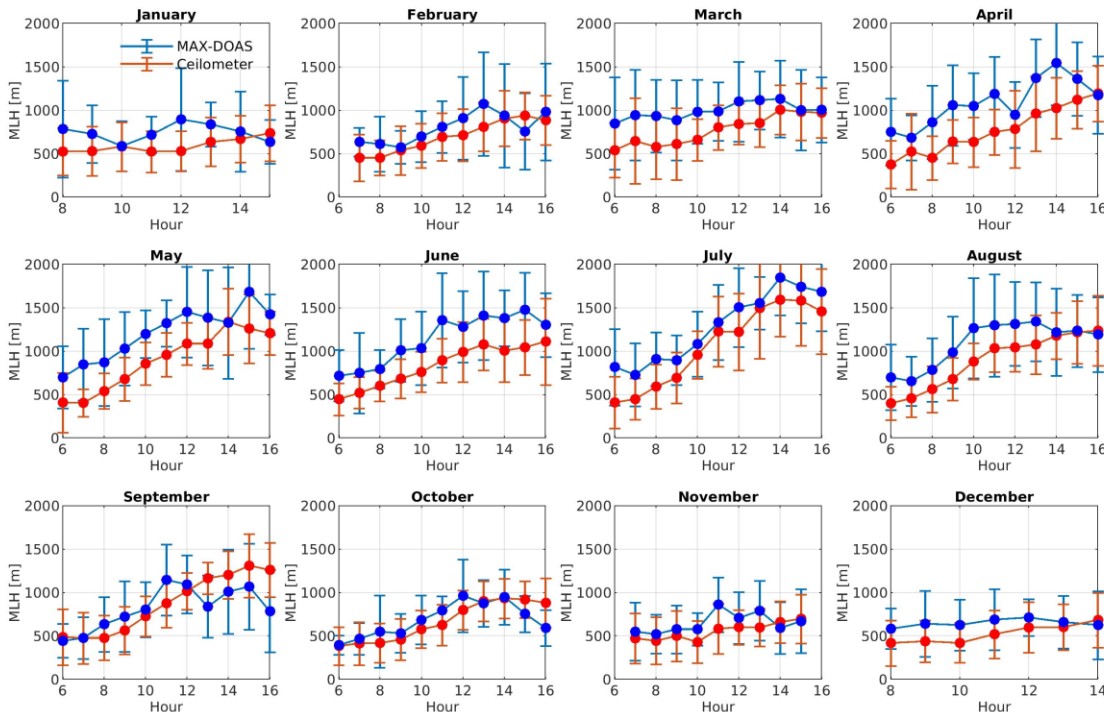

**Figure 3: Comparison of monthly averaged MLH diurnal variations as estimated by the BIRA-IASB MAX-DOAS measurements and the co-located ceilometer. The error bars for both datasets represent the standard deviation ($\pm 1\sigma$) of the hourly mean per month.**





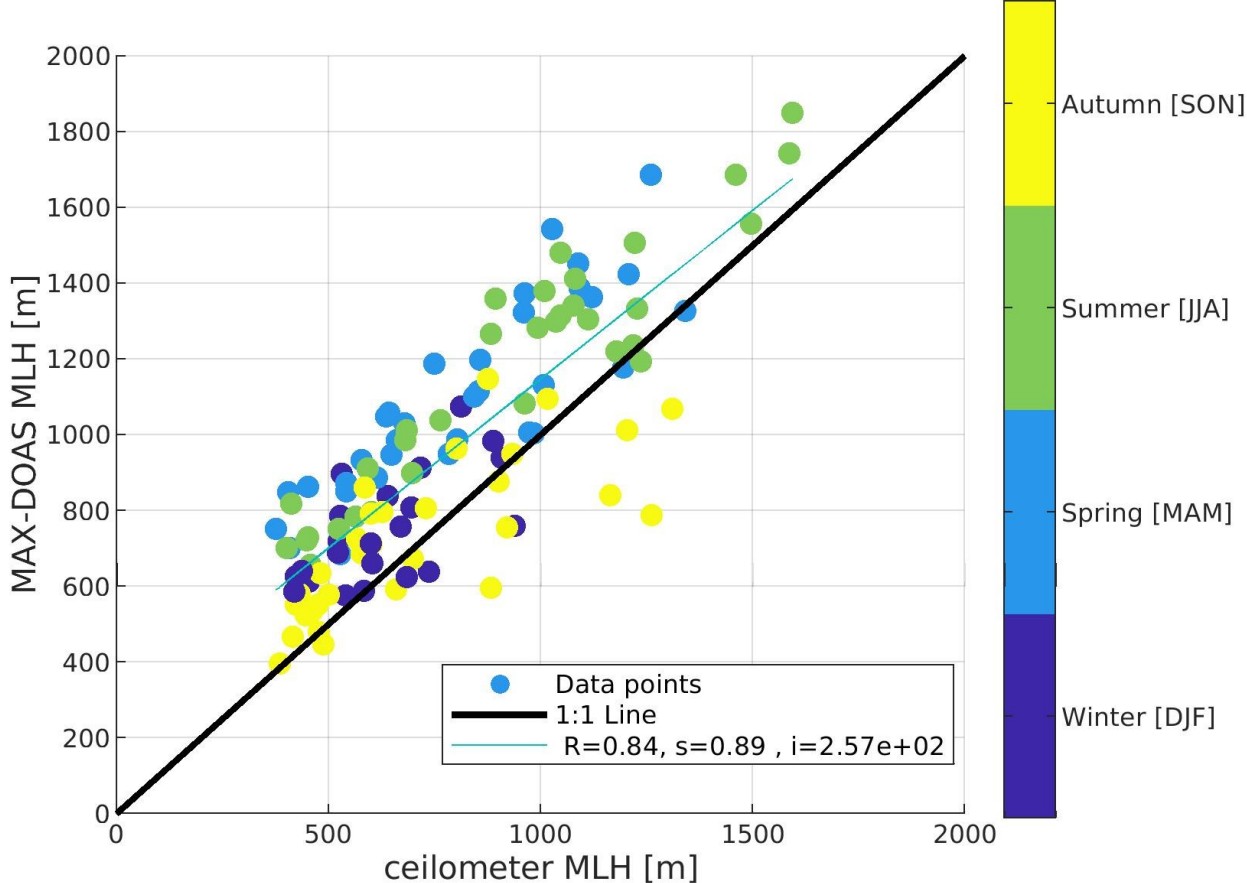

**Figure 4: Scatter plot of the monthly average MLH diurnal variation values of the MAX-DOAS and the co-located ceilometer. The color bar separates the data by season.**





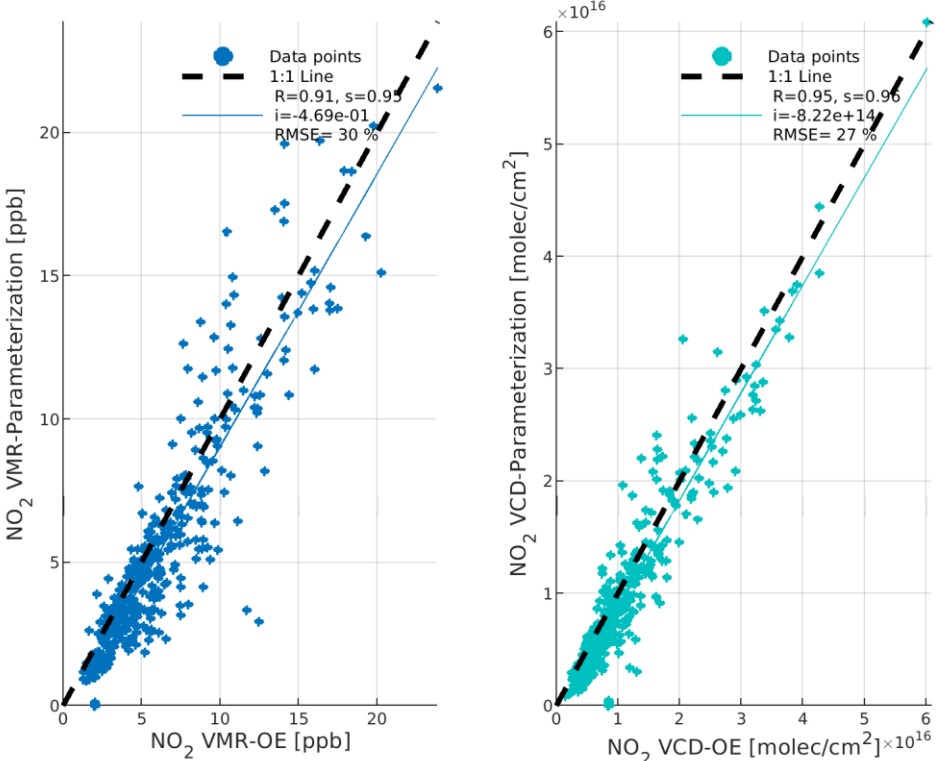

**Figure 5: VIS Range: Comparison between (left panel) MMF and parameterized NO₂ VMR and (right panel) MMF and parameterized NO₂ VCD.**





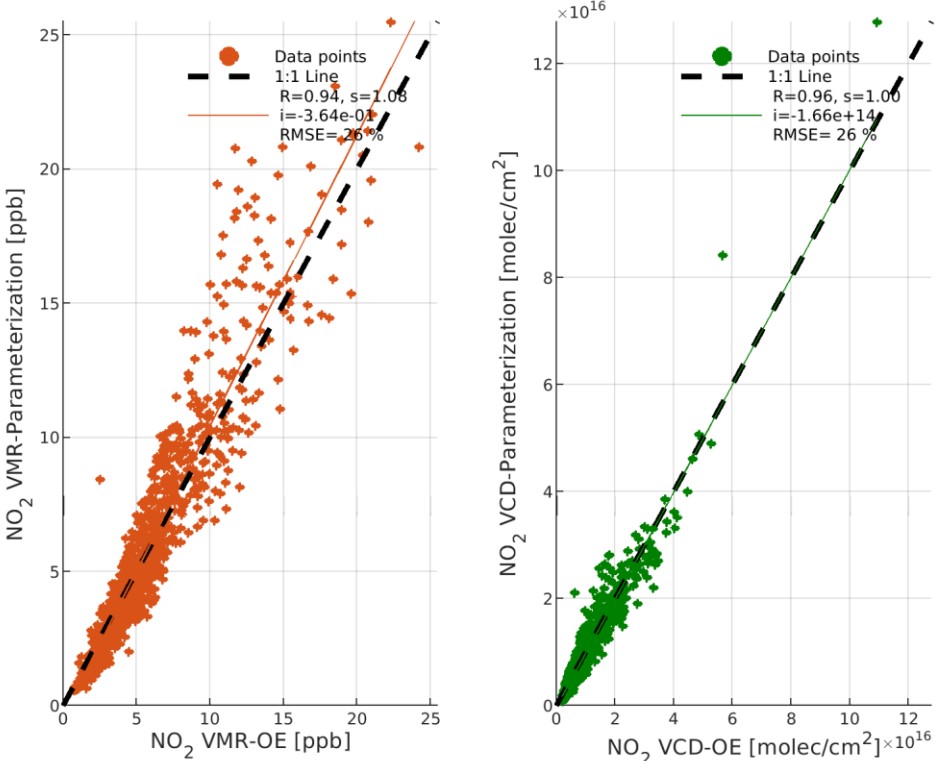

**Figure 6: Same as Figure 5 for the UV channel.**

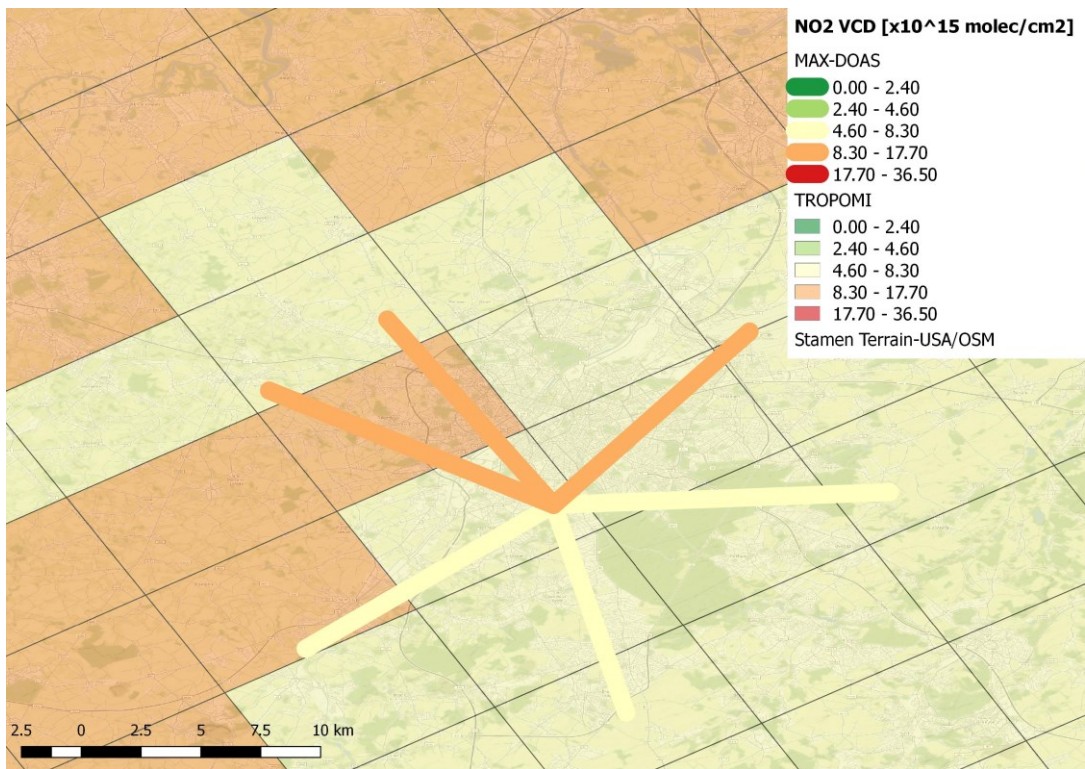

**Figure 7: Tropospheric NO₂ columns derived from the TROPOMI (pixel size equal to 7 x 3.5 km²) and the MAX-DOAS instrument on 06 June 2018 near the measurement site in Uccle (overlaid onto OSM Standard layer). © OpenStreetMap contributors 2020. Distributed under a Creative Commons BY-SA License.**





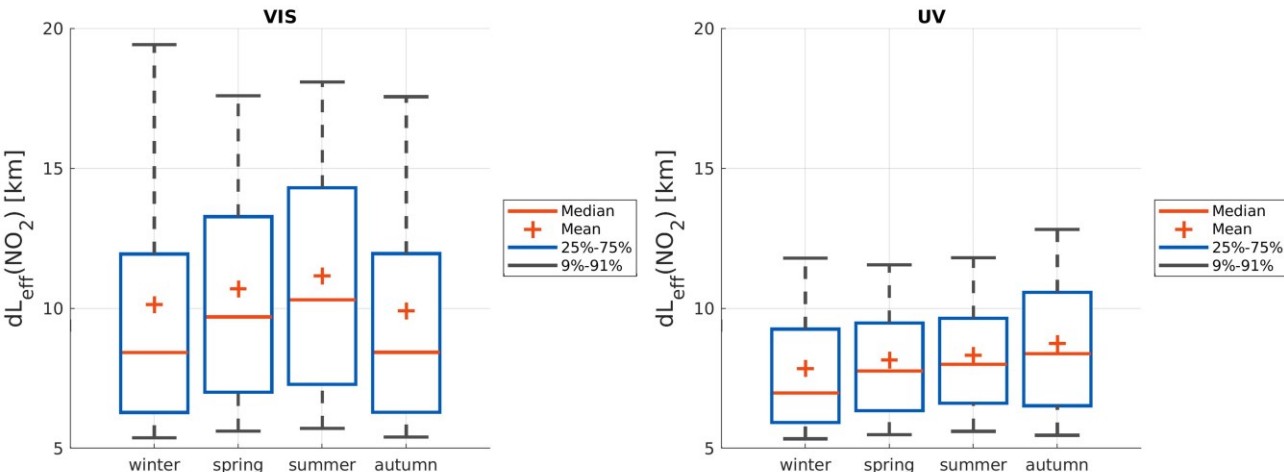

**Figure 8: Box and whisker plots representing the seasonal horizontal sensitivity as derived from all the azimuthal viewing directions by applying the parameterization method for the Vis (left panel) and UV (right panel) spectral ranges. Every seasonal box contains the estimated dLeff (NO2) of all the azimuthal viewing directions.**







**Figure 9: Seasonally-averaged near-surface NO₂ VMR around 11 UTC as a function of azimuthal viewing direction derived by the parameterization technique in the Vis and UV wavelength ranges. Lines with black borders represent the UV VMRs, and lines without black borders show the Vis VMRs. The length of each line represents the seasonally-averaged horizontal sensitivity. Different color scales are used per season. © OpenStreetMap contributors 2019. Distributed under a Creative Commons BY-SA License.**

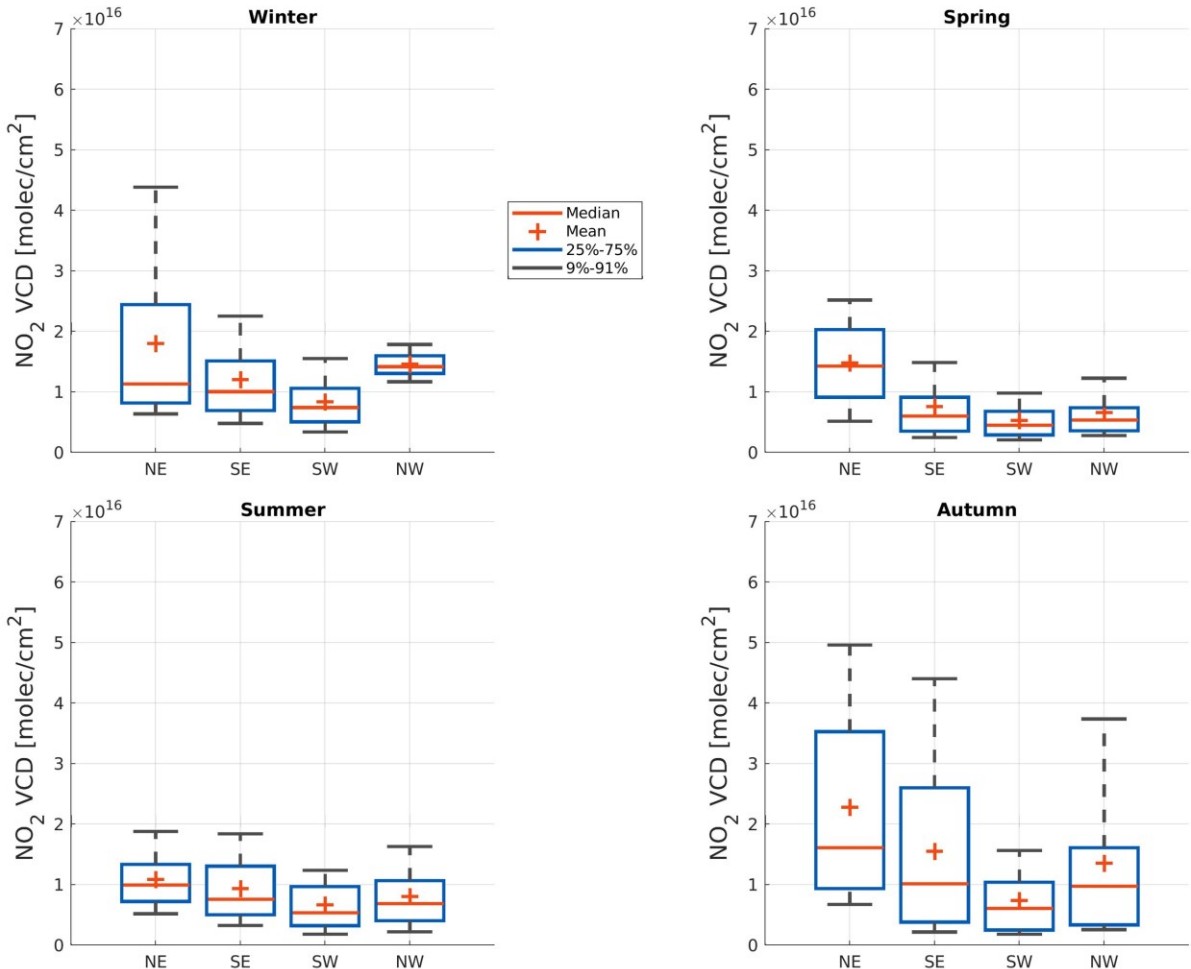

**Figure 10: Box and whisker plots representing, for each season, the tropospheric NO₂ column as a function of the wind direction. The MAX-DOAS columns are derived by the parameterization technique in the Vis wavelength range (all the azimuthal directions are included) and the wind observations from the meteorological station on the BIRA-IASB rooftop.**




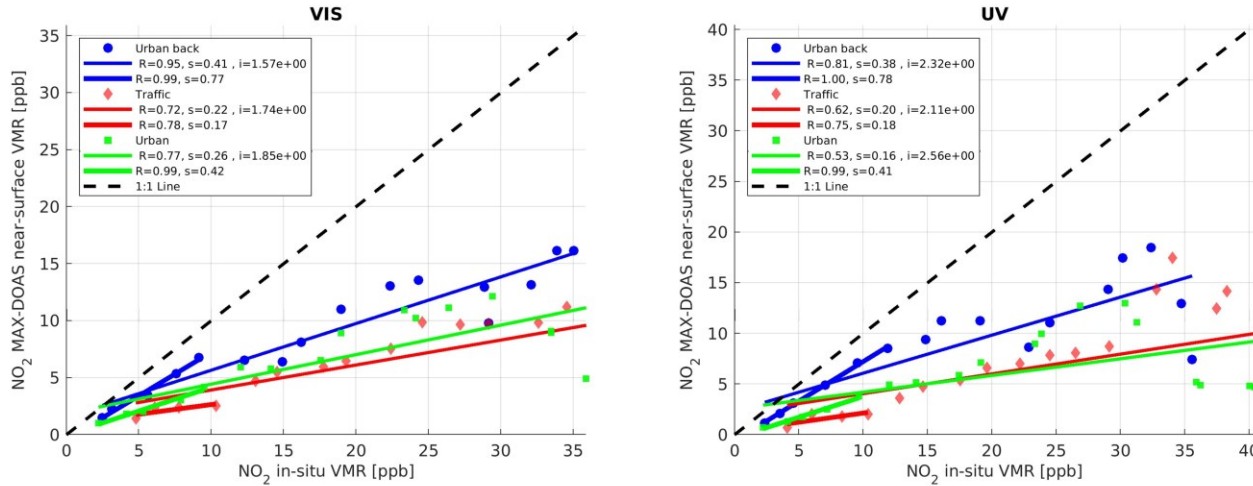

**Figure 11: Scatter plots of binned MAX-DOAS and in-situ NO₂ near-surface VMR in the (left panel) VIS and (right panel) UV channels. The solid thick lines are the regression analysis results for the 0-12 ppb in-situ NO₂ concentration range.**



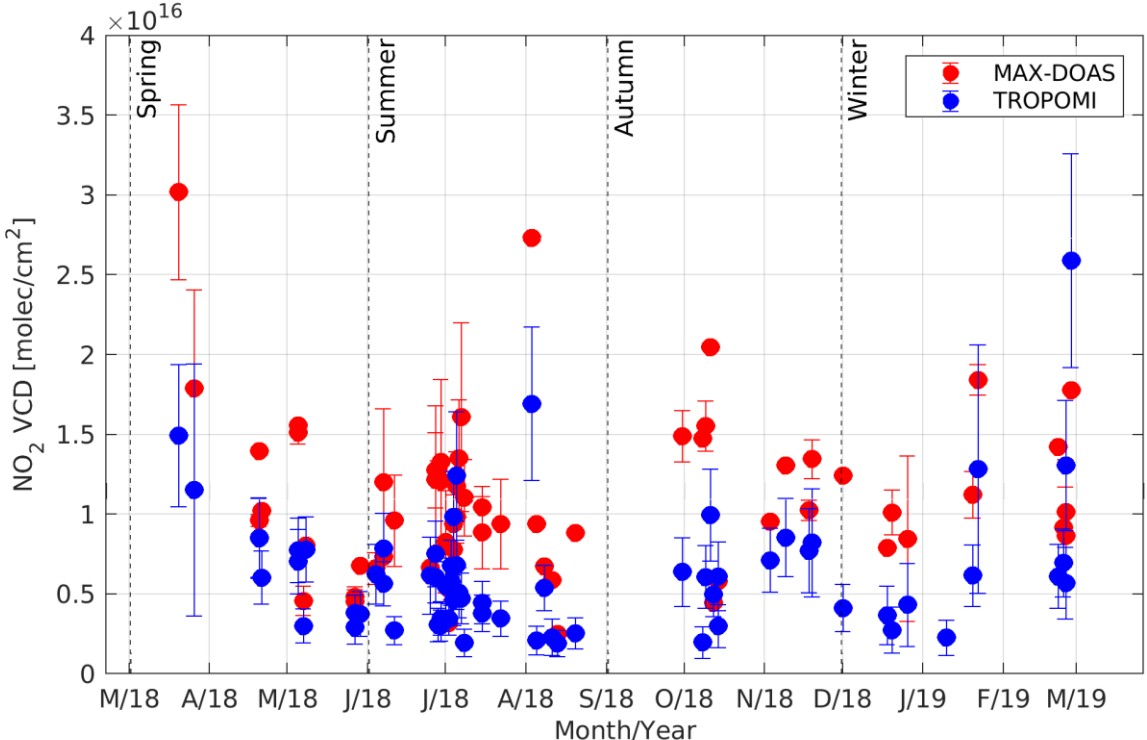

**Figure 12: Time series of the tropospheric NO₂ columns derived from the main azimuthal MAX-DOAS direction observations in the Vis range and the closest TROPOMI pixel located along the MAX-DOAS azimuthal direction. The MAX-DOAS error bars represent the standard deviation of the averaged values within one hour before and after TROPOMI's overpass time while the TROPOMI error bars are equal to the TROPOMI VCD error as provided in the data files.**



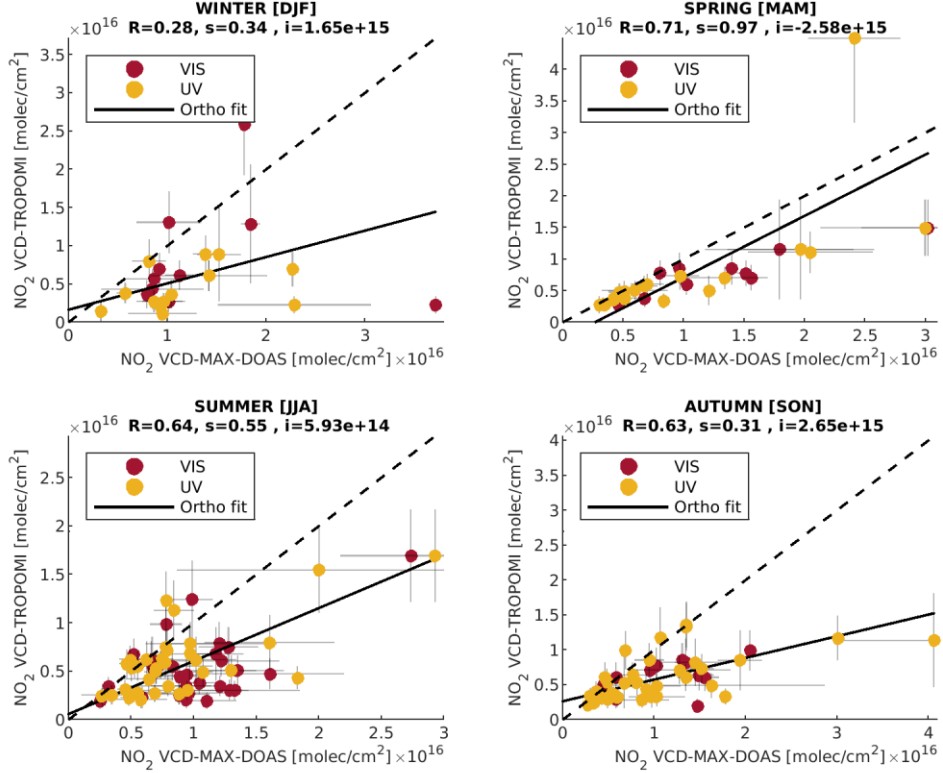

**Figure 13: Seasonal scatter plots between the tropospheric NO₂ columns derived by UV and Vis MAX-DOAS observations (yellow and red circles, respectively) in the main azimuthal direction and the closest TROPOMI pixel with respect to the measurement site. The slope is estimated by using orthogonal regression analysis. The error bars are the same as in Figure 11.**


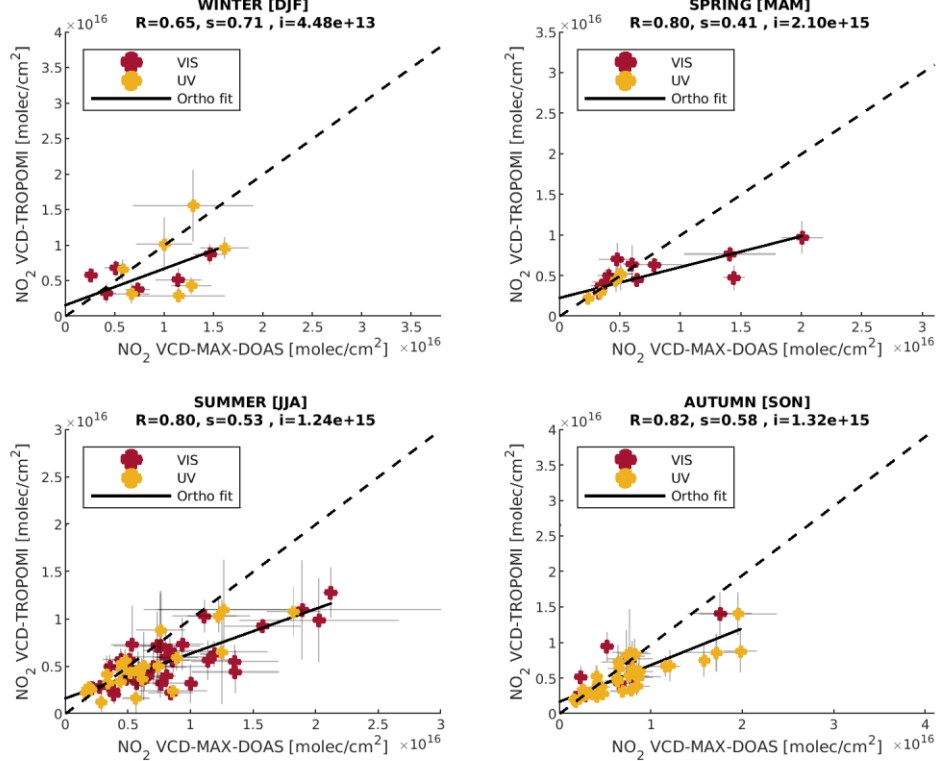

**Figure 14: Seasonal scatter plots between the tropospheric NO₂ columns derived from the dual-scan MAX-DOAS observations (in the Vis and UV together) and the TROPOMI pixels by using information about the MAX-DOAS horizontal effective light path and the co-location between pixels and azimuthal directions. The MAX-DOAS error bars are the same as in Figure 11. The TROPOMI error bars represent the standard deviation of the averaged pixel values within a circle with radius equal to the MAX-DOAS horizontal effective light path.**

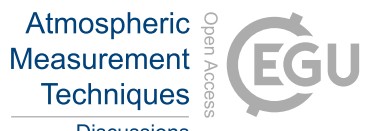

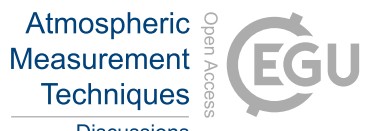

**Figure 15: Mean TROPOMI averaging kernels (blue line) and median MAX-DOAS NO₂ profile during summer as a function of altitude.**



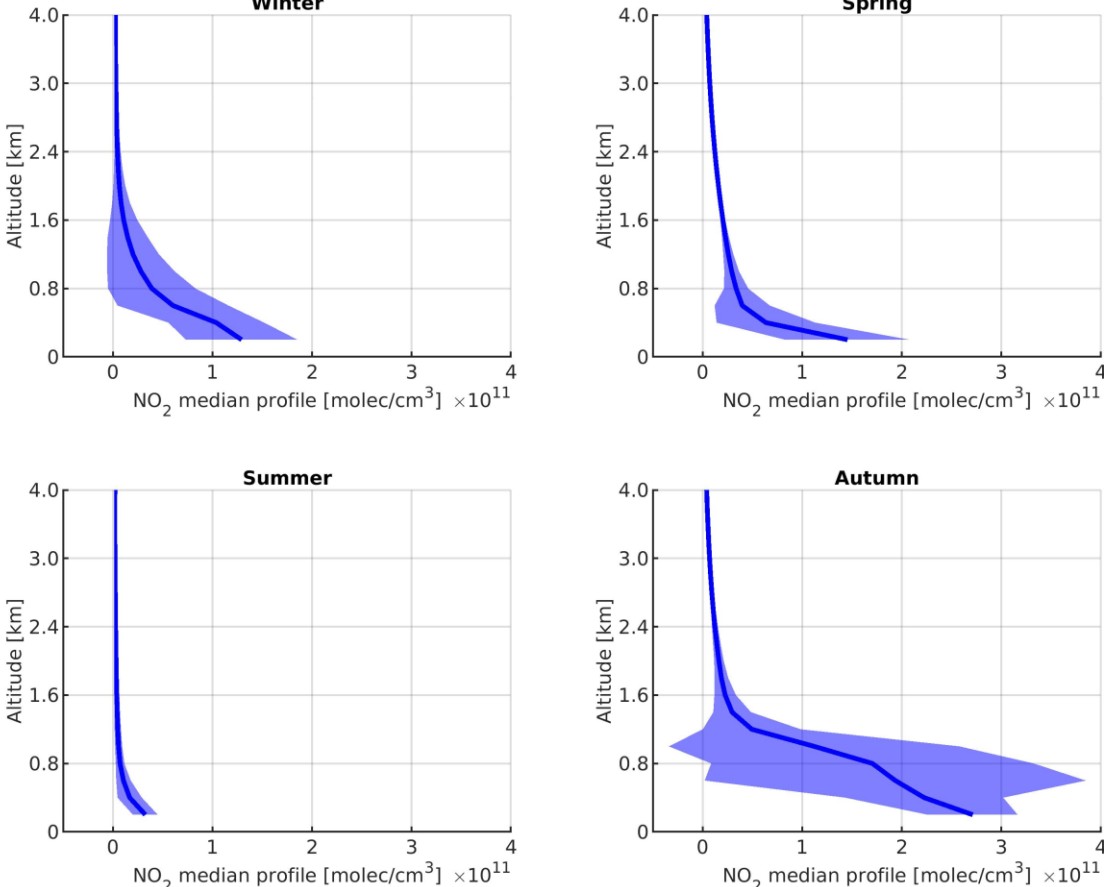

**Figure 16: Example of median daily MAX-DOAS Vis NO₂ profiles, which are used for the recalculation of the TROPOMI NO₂ columns (one example day per season: 13 May 2018, 12 August 2018, 29 October 2018 and 19 January 2019). The shaded areas ($\pm 1\ \sigma$) represent the variability of the daily profiles.**

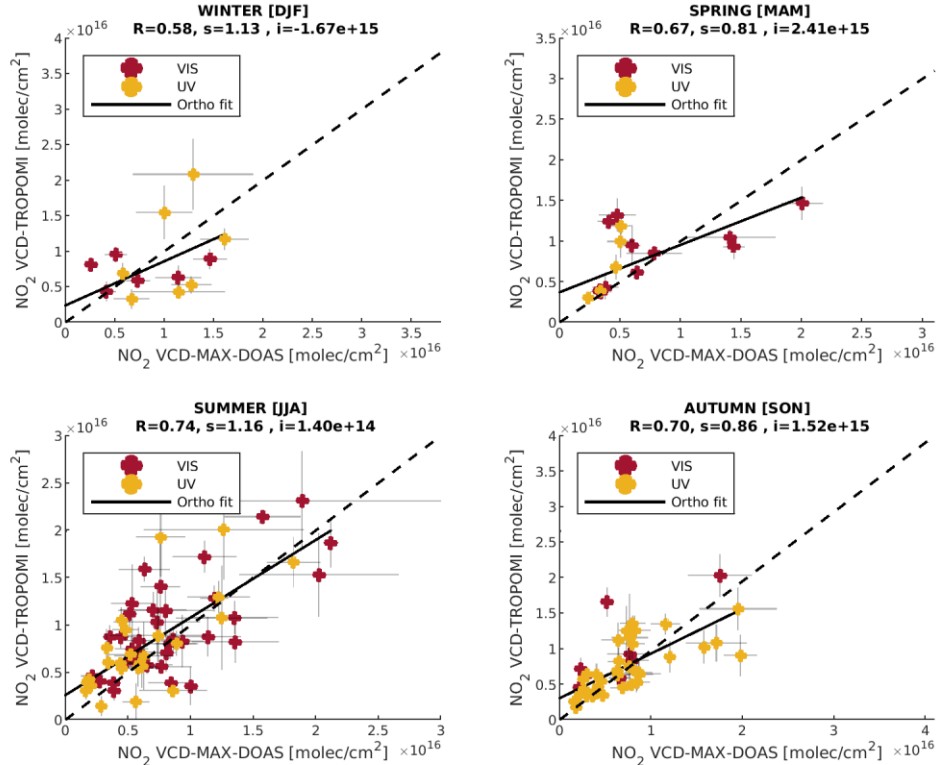

**Figure 17: Seasonal scatter plots between the tropospheric NO₂ columns derived from the 2-D MAX-DOAS UV and VIS observations and TROPOMI NO₂ columns recalculated using the median daily MAX-DOAS vertical profiles as a-priori information.**

