# Peer review of "Validation of TROPOMI tropospheric NO2 columns using dual-scan MAX-DOAS measurements in Uccle, Brussels"

_Atmospheric Measurement Techniques, 2020_

## Referee Comment (RC1) · Anonymous Referee #1 · 2 Apr 2020

The manuscript with the title "Validation of TROPOMI tropospheric NO2 columns using dual-scan MAX-DOAS measurements in Uccle, Brussels" (1) represents a substantial contribution to scientific progress in validating TROPOMI tropospheric NO2 column observations. (2) Scientific approaches and applied methods are valid and results are discussed in an appropriate way. (3) Scientific results and conclusions are presented in a clear and well structured way.

I strongly recommend the publication of this manuscript, after consideration of a number of specific comments.

Specific comments:

[Figure]

- Could you add MetOp-C as well? (Page 2, Line 14)

- Could you give some numbers? What are the European standards? On how many days the standards are exceeded at this station? (Page 3, Line 20)

- Is this number correct? In Kreher et al. (2019) you mention -50°C. Please check again. (Page 4, Line 5)

- Could you give some details on MMF performance? What are its strengths/weakness compared to other algorithms? (Page 5, Line 13)

- Is there any reasons why you don't use data from radiosondes instead of standard profiles? Is there any nearby station? (Page 5, Line 18)

- How far is the AERONET station away from the MAX-DOAS station? (Page 5, Line 22)

- Is the temperature dependence on NO2 cross section relevant for UV and Vis, or only for Vis? (Page 6, Line 23)

- Does the direct use of dLeff only lead to underestimation of NO2 VMR? A factor of three appears quite large. Is a factor of three appropriate for such low elevation angle as used in your study? (Page 7, Line 16)

- Previous studies have highlighted the importance of properly estimating correction factors, but did any of these studies compare direct use of dLeff (dLeffO4, without fc) with corrected dLeff (dLeffNO2? I would suggest to also compute dLeffO4 (e.g. Seyler et al. 2018) and compare with dLeffNO2. I would be surprised to see a factor of three difference. I suggest to add one plot (dLeffO4 for UV and Vis) to Fig. 8 (b). This would really help to know how essential are such correction factors for urban settings and low elevation angles. (Page 7, Line 26)

- Did you take AOD, asymmetry parameter, and SSA values from the AERONET station? AOD at which wavelength?

- Why did you not use data from weather station for the calculation of nair? (Page 9, Line 7)

- Did you include uncertainties arising from the use of AFGL profiles instead of data from weather stations? (Page 9, Line 25).

- Throughout the whole manuscript you are using the terms MLHNO2 and MLHMAX-DOAS but actually, as I understand, the two terms refer to the same parameter? I suggest to use only MLHMAXDOAS?

- How far is the ceilometer away from the MAX-DOAS station? (Page 11, Line 10). I suggest to include the position of AERONET and ceilometer stations in Fig. 1, if the position is other than for the MAX-DOAS instrument.

- Can you give some details on cloud screening? Did you use cloud-screened pixels only? (Page 12, Line 16)

- Please also include a few sentences discussion about dLeffO4 and compare with dLeffNO2. (Sect. 4.1, Fig. 8).

- Did you use 11:00 UTC because of TROPOMI overpass? If so, please add this information. (Page 13, Line 5)

- Is it really only up to 200 m? I would suppose values up to 350 m for dLeff = 10 km and EA = $2°$ and also I expect differences for UV and Vis channels, according to Wang et al. 2014 AMT. (Page 15, Line 6).

- Because you mention that one azimuthal MAX-DOAS measurement samples air masses along several kilometers, what about the correlation between MAX-DOAS (geometric approach, e.g. using $30°$ measurements) and TROPOMI? (Page 16, Line 17)

- Actually I do not see improvement for summer. (Page 17, Line 4)

- Again, not clear which cloud fraction you used for cloud screening. (Page 17, Line 19)

- Is it mean or median? (Page 18, Line 30) Because in the conclusion you state that you are using median MAX-DOAS profiles as a priori. (Page 20, Line 17)

Technical corrections:

- associated with (Page 11, Line 29)

- Two modifications are introduced (Page 16, Line 24)

- Some words (e.g. . . . algorithm is based on NO2 . . .) are missing in the first sentence of Sect. 4.4.3 (Page 18, Line 19)

---

## Referee Comment (RC2) · Anonymous Referee #2 · 22 May 2020

The manuscript presents the results of MAX-DOAS measurements for a combination of vertical and azimuthal scans, allowing for the investigation of horizontal gradients in NO2 concentrations, and demonstrates that the validation of TROPOMI columns benefits from this extra information. The paper is well written and matches to the scope of AMT. It should be published after dealing with the comments below.

Major drawback: Standard atmosphere

The authors use a standard atmosphere for their evaluation. This is quite odd, as the MAXDOAS measurements are taken right at the meteorological institute of Belgium, where I would expect that atmospheric profiles of temperature and pressure would be

available on daily basis.

The authors discuss the seasonality of their results. This discussion is hard to assess as the seasonality of T and p, which directly affects the O4 concentration (and thus aerosol inversion and thus NO2), is just ignored. Thus the authors at least have to study the impact of the used standard atmosphere on the seasonality of results. But I would highly recommend that the authors redo the complete data analysis for real atmospheric profiles.

Comments

5/25: For a DoF of 1, there is only 1 piece of information available (i.e. the total column). So for real "profile information" I would expect a threshold of DoF>2 rather than 1.

6/3: Is the pyrometer also pointing in the same direction as the MAX-DOAS instruments, or does it have a fixed viewing direction? If the latter, how could it provide information on cloudiness for low elevetaion angles?

6/27: Please add a direct comparison of the NO2 results for UV and Vis and compare with the listed uncertainties.

6/29: The subsection heading announces details on the dual scan retrieval. However, large parts of the following text just explain the par method. I propose to add a subsection dedicated to the par method for better clarity.

16/4: Please put this finding in relation to other comparisons (which have also reported a low bias of TROPOMI columns) and add respective references.

17/26: I don't agree with the general statement that cloud effects are quasi random and do not cause systematic biases, and I don't see these bold statements supported by the presented measurements. Cloud impact is definitely "complex" and may indeed "lead to positive or negative biases". So they definitely introduce considerable scatter to the retrieved columns. But underneeth this scatter, which might look "quasi random", there are very likely systematic effects as well, which probably could only be quantified

with large data samples.

Table 1: Please add the different sets of azimuth angles to Fig. 1, color coded for the three time periods.

Table 4: Please add results from a direct comparison between NO2 UV and vis - is the difference within the listed total uncertainty range?

Fig. 1: Please add the different sets of azimuth angles, color coded for the three time periods. What is the meaning of the symbols on top of the azimuth direction lines?

Fig. 7: The landscape background is not helpful here, but rather disturbing. I propose to show no background, but instead real color coded VCDs for both TROPOMI and MAXDOAS.

Fig. 12: Several MAX-DOAS VCDs have no errorbar. I assume that for these days no std could be calculated. I propose to also show the mean/typical uncertainty of single VCDs derived from the MAX-DOAS inversion, which would probably be more consistent for the time period. This might be added as second error bar with e.g. light color.

Fig. 15/16: Please add the corresponding a-priori profiles used in the TROPOMI retrieval, and add a discussion of this comparison.

Minor comments:

1/18: "concentrations" should be "volume mixing ratios"

3/26: "eventual" should be "potential" or "possible"

6/2: Skip the comma between "measurements" and "strongly"

6/10: "Smoothing error"

6/30: Please define what exactly is meant by "near surface VMR"

8/29: "are equal" should be "are similar"

9/21: "close to zero" is not unphysical

---

## Author Comment (AC2) · 26 Jun 2020

The comment was uploaded in the form of a supplement:
https://www.atmos-meas-tech-discuss.net/amt-2020-33/amt-2020-33-AC2-supplement.pdf

---

## Author Response (AR1)

Dear Editor,

With regard to the manuscript:
Amt-2020-33
5   Title: Validation of TROPOMI tropospheric $NO_2$ columns using dual-scan MAX-DOAS measurements in Uccle, Brussels

Author(s): E. Dimitropoulou et al.

10  Please find below the changes made in the revised manuscript according to the referees' comments. Please consider that:
    A) Green bold: Comments of the Referee
    B) Black bold: The response to each referee's comment
    C) Red bold: Added text in the manuscript according to referee's comments.

Sincerely yours,

E. Dimitropoulou (ermioni.dimitropoulou@aeronomie.be)

20  **Response to Anonymous Referee #1**

**1. Comment: Could you add MetOp-C as well? (Page 2, Line 14)**
**Response:** The MetOp-C satellite is now added to the text (Page 2, Line 14) as follows:
"Those observations started in 1995 with the ERS-2 GOME (Global Ozone Monitoring
25  Experiment) instrument (Burrows et al., 1999), followed in chronological order by ENVISAT-SCIAMACHY (SCanning Imaging Absorption spectroMeter for Atmospheric CHartographY) in 2002 (Bovensmann et al., 1999), AURA OMI (Ozone Monitoring Experiment) in 2004 (Levelt et al., 2006), MetOp-A/ GOME-2A in 2006, MetOp-B/ GOME-2B in 2012 **and MetOp-C in 2018** (Munro et al., 2016)."

**2. Comment: Could you give some numbers? What are the European standards? On how many days the standards are exceeded at this station? (Page 3, Line 20)**

**Response:** The following paragraph is added (Page 3, Line 21), which quotes the European
35  standards for the NO2 values and what is happening in Brussels according to annual reports from the Bruxelles Environment/ Leefmilieu Brussel agency:

"Frequently, the $NO_2$ concentration monitored by the network of telemetric air quality stations from Bruxelles Environment/Leefmilieu Brussel (https://environnement.brussels/) often
40  exceeds the European standards **upper limits fixed to 40 and 200 μg/m$^3$ for the $NO_2$ annual and hourly mean concentrations, respectively. For instance in 2015, the NO$_2$**

**annual mean concentrations have been found to exceed those European standards in narrow busy streets in Brussels with an annual mean concentration between 42.5 to 52.5 μg/m³ (see https://environnement.brussels/). Additionally, as detected by satellite sensors like OMI (Huijnen et al., 2010), NO₂ columns over Brussels are among the highest in Europe."**

**3. Comment: Is this number correct? In Kreher et al. (2019) you mention -50°C. Please check again. (Page 4, Line 5)**

**Response:** The remark is correct and the temperature has been corrected to 223 K (Page 4, Line 9).

**4. Comment: Could you give some details on MMF performance? What are its strengths/weakness compared to other algorithms? (Page 5, Line 13)**

**Response:** The following paragraph, which describes the strengths/weakness of MMF inversion algorithm, has been added to the manuscript, (Page 5, Line 15:20):
**"The main advantages of MMF are (1) the on-line calculation of Jacobians with operation in a logarithmic state vector space, which prevents unphysical negative partial columns to be retrieved, (2) the use of a stable Levenberg-Marquardt non-linear iteration scheme (in replacement to a Gauss Newton scheme) and (3) the fast computing time (5 s per scan for both aerosols and trace gases). A drawback of OEM-based profiling algorithms such as MMF, is that a priori profiles should be carefully chosen in order to avoid biases in the retrieved profiles and columns at altitudes characterized by a low information content."**

**5. Comment: Is there any reasons why you don't use data from radiosondes instead of standard profiles? Is there any nearby station? (Page 5, Line 18)**

**Response:** When the study started, the pressure and temperature profiles from the US Standard Atmosphere were used because they are easily accessible. However it is true that the Royal Meteorological Institute of Belgium is performing radiosondes and the use of such T/p profiles would be more appropriate as we investigate the seasonality of the MAX-DOAS retrievals. Given the fact that this point has been raised by both reviewers, we decided to redo the complete data analysis using 20-year-based monthly averages of T/p profiles from the ECMWF ERA Interim reanalysis output extracted for Brussels, Uccle. The following sentences have been modified in the manuscript:

"The pressure and temperature profiles are taken from the Air Force Geophysics Lab (AFGL) 1976 Standard Atmosphere (Anderson et al., 1986). The variation of the temperature profiles

during one year of measurements is taken into account as an additional error on the profile retrieval."

now becomes:

"**The pressure and temperature profiles are prescribed using 20-year-based monthly averaged data extracted from the European Centre for Medium-Range Weather Forecasts (ECMWF) ERA Interim reanalysis (see Beirle et al., 2019) for the location of Uccle.**" (see page 5, Line 26)

and

"The near-surface VMR is then obtained by dividing the concentration of the trace gas (Eq. 4) by the air number density ($n_{air}$). For the calculation of $n_{air}$, the pressure and temperature profiles were taken from the AFGL 1976 Standard Atmosphere (Anderson et al., 1986) and are the same as used in Section 2.3.1."

is now:

"The near-surface VMR is then obtained by dividing the concentration of the trace gas (Eq. 4) by the air number density ($n_{air}$) **derived from monthly averaged temperature and pressure profiles extracted from the ERA Interim reanalysis (see Section 2.3.1)**." (see page 9, Line 24)

**6. Comment: How far is the AERONET station away from the MAX-DOAS station? (Page 5, Line 22)**

**Response:** The AERONET station is located on the rooftop of the BIRA-IASB building, at a distance of 180 m from the MAX-DOAS station. The location of the AERONET station as well as the ceilometer are now added to Figure 1.

**7. Comment: Is the temperature dependence on NO2 cross section relevant for UV and Vis, or only for Vis? (Page 6, Line 23)**

**Response:** The temperature dependence as adopted by Takashima et al. (2012) is relevant for the Vis fitting window. Although it is not strictly applicable to the UV range, the temperature dependence of the $NO_2$ cross section is similar in both spectral regions. Therefore, we used the same uncertainty estimate for both fitting windows.

**8. Comment: Does the direct use of dLeff only lead to underestimation of NO2 VMR? A factor of three appears quite large. Is a factor of three appropriate for such low elevation angle as used in your study? (Page 7, Line 16)**

**Response:** According to Sinreich et al. (2013), Ortega et al. (2015) and Figure 2 in our study, when the correction factors are larger than one, the method becomes highly dependent on the aerosol load. So, these measurements are excluded from the data analysis. Based on that, when applying the appropriate correction factors, the dLeff_NO2 can only be underestimated and consequently, the NO$_2$ VMR can only be overestimated. When comparing NO$_2$ VMRs before and after the use of correction factors, they differ by a mean factor of 2 and 1.6 for Vis and UV, respectively. The following sentence has been added on Page 8, Line 1-2:

"**In the present study, where off-axis measurements were performed at 2° elevation in an urban polluted environment, the dL$_{eff}$ of NO$_2$ was found to be smaller than the corresponding dL$_{eff}$\_O$_4$ by an average factor of 2 and 1.6, in the Vis and UV wavelength ranges, respectively (Fig. 8).**"

**9. Comment: Previous studies have highlighted the importance of properly estimating correction factors, but did any of these studies compare direct use of dLeff (dLeffO4, without fc) with corrected dLeff (dLeffNO2? I would suggest to also compute dLeffO4 (e.g. Seyler et al. 2018) and compare with dLeffNO2. I would be surprised to see a factor of three difference. I suggest to add one plot (dLeffO4 for UV and Vis) to Fig. 8 (b). This would really help to know how essential are such correction factors for urban settings and low elevation angles. (Page 7, Line 26)**

**Response:** For the first part of your comment, please see our response to your previous comment. The figure, which shows the dLeffO4 for Vis and UV, has been added to Figure 8:

[Figure]

As we can see, the difference between the two dLeff (before and after the use of appropriate correction factors) can become quite large and is approximately about a mean factor of 2 and 1.6 for Vis and UV, respectively (Page 13, Line 14). For extreme cases, the difference between the two dLeff can reach a factor of 5 and 4 for Vis and UV, respectively. Consequently, the use of appropriate correction factors under urban conditions and measurements at low elevation angles is critical. The following paragraph has now been added in the text:

"**Similarly, dLeff(O$_4$), which represents the horizontal sensitivity before applying the appropriate correction factors, is larger in the Vis than in the UV range. The dLeff(O$_4$) Vis can reach values of up to 28 km, while the maximum value for UV is around 18 km. The difference between dL$_{eff}$ (NO$_2$) and dLeff(O$_4$) can become quite large reaching approximately a factor of 2 and 1.6 for Vis and UV, respectively. For extreme cases, this difference factor reach up to 5 and 4 for Vis and UV, respectively.**" (see Page 13, Line 16).

**10. Comment: Did you take AOD, asymmetry parameter, and SSA values from the AERONET station? AOD at which wavelength?**

**Response:** The page and line of your comment is not indicated but we assume that it is for the correction factors estimation. As indicated in Sinreich et al. (2013), appropriate correction factors (that are not dependent on AOD) can be calculated when a certain aerosol load is

reached in the atmosphere. This aerosol load is reached for AOD in the range of 0.3 - 0.6. In the present study, as indicated in the manuscript, the dAMFs for AOD equal to 0.3 are used for the analysis.

**11. Comment: Why did you not use data from weather station for the calculation of nair? (Page 9, Line 7)**

**Response:** As mentioned above, the data analysis was re-done replacing the AFGL 1976 Standard Atmosphere by more realistic profiles obtained using 20-year-based monthly averages of temperature/pressure profiles from the ECMWF ERA Interim reanalysis extracted for Uccle, Brussels. The text in the manuscript has been adapted accordingly (see our reply to your comment n°5 above).

All the figures and discussion results have been adjusted to the new data analysis results. When comparing the $NO_2$ VCDs, $NO_2$ near-surface VMRs and $MLH(NO_2)$ derived using the revised atmospheric profiles with those obtained with the initial climatology, we report the following changes:

| Value | Percentage difference |
|---|---|
| **NO2 VCD Vis** | Winter: -7.1 %
 Spring : -7.7 %
 Summer: -8.5 %
 Autumn: - 6.5 % |
| **NO2 VCD UV** | Winter: -7.2 %
 Spring : -9.2 %
 Summer: -10.2 %
 Autumn: -8.6 % |
| **NO2 VMR Vis** | Winter: -1.8 %
 Spring : -5.4 %
 Summer: -9.6 %
 Autumn: -5.4 % |
| **NO2 VMR UV** | Winter: -0.9 %
 Spring : -5.3 %
 Summer: -10.1%
 Autumn: -6.1 % |
| **AOD Vis** | Winter: 0.6 %
 Spring : -2.5 %
 Summer: -6.3 %
 Autumn: -5.8 % |
| **AOD UV** | Winter: 2. 9 %
 Spring : -2.5 %
 Summer: - 8.0 %
 Autumn: -6.0 % |
| **MLH (NO2)** | Winter: -5.9 % |

| | Spring : -2.7 % |
| | Summer: 0.5 % |
| | Autumn: -1.7 % |

The percent differences are calculated against the initial data (AFGL 1976 Standard Atmosphere).

**12. Comment: Did you include uncertainties arising from the use of AFGL profiles instead of data from weather stations? (Page 9, Line 25).**

**Response:** As mentioned above, the whole analysis was redone using climatological temperature/pressure profiles based on ECMWF reanalysis data. An estimation of the impact on NO2 near-surface VMR and VCD when using the ECMWF climatology instead of US standard atmosphere is presented in the previous comment n° 11. Unfortunately, temperature/pressure profiles from weather stations were not easily accessible so, these uncertainties could not be estimated.

**13. Comment: Throughout the whole manuscript you are using the terms MLHNO2 and MLHMAXDOAS but actually, as I understand, the two terms refer to the same parameter? I suggest to use only MLHMAXDOAS?**

**Response:** The two terms (MLH_NO2 and MLH_MAXDOAS) refer to same parameter. The term MLH_MAXDOAS has been used in the manuscript.

**14. Comment: How far is the ceilometer away from the MAX-DOAS station? (Page 11, Line 10). I suggest to include the position of AERONET and ceilometer stations in Fig. 1, if the position is other than for the MAX-DOAS instrument.**

**Response:** The AERONET instrument, as mentioned above, is located 180 m away from the MAX-DOAS spectrometer and the ceilometer is located 160 m away from the MAX-DOAS location. The locations of both AERONET and ceilometer instruments are now included in Figure 1 (right panel):

[Figure]

**15. Comment: Can you give some details on cloud screening? Did you use cloud-screened pixels only? (Page 12, Line 16)**

**Response:** As clouds in the atmosphere can strongly affect the MAX-DOAS trace gas retrieval, a cloud filtering approach is applied to the MAX-DOAS scans by using a co-located thermal infrared pyrometer. In the present study, only MAX-DOAS scans with a total cloud-cover fraction less than 0.8 (80 %) are selected for further analysis. By applying this upper
10 limit, scans under fully cloudy conditions are rejected (Page 6, Line 9).

Concerning the TROPOMI cloud filtering, in the present study, only pixels with a quality assurance value larger than 0.75 (QA>0.75) are included in the validation. This filter removes pixels with a cloud radiance fraction larger than 0.5. This has been added in the text (Page 12, Line 32).

**16. Comment: Please also include a few sentences discussion about dLeffO4 and compare with dLeffNO2. (Sect. 4.1, Fig. 8).**

**Response:** After comparing dLeffO4 with dLeffNO2, the following paragraph has now been
20 added in Section 4.1 (see our reply to comment n°9 above):

"**Similarly, dLeff($O_4$), which is the horizontal sensitivity before applying the appropriate correction factors, is larger in the Vis than in the UV range. The dLeff($O_4$) Vis can reach values of up to 28 km, while the maximum value for UV is around 18 km. The difference**
25 **between $dL_{eff}$ ($NO_2$) and dLeff($O_4$) can become quite large by approximately a mean factor of 2 and 1.6 for Vis and UV, respectively. For extreme cases, this difference factor can be up to 5 and 4 for Vis and UV, respectively.**" (see page 13, line16).

**17. Comment: Did you use 11:00 UTC because of TROPOMI overpass? If so, please add this information. (Page 13, Line 5)**

**Response:** The TROPOMI overpass over Brussels is between 10:00 and 12:00 UTC. For that reason, we have chosen to present the MAX-DOAS results at 11:00 UTC. The following sentence is now added in Page 13, Line 22:

"**The choice of presenting the MAX-DOAS measurements at 11:00 UTC is based on the TROPOMI overpass time, which varies between 10:00 and 12:00 UTC.**"

**18. Comment: Is it really only up to 200 m? I would suppose values up to 350 m for dLeff = 10 km and EA = 2_ and also I expect differences for UV and Vis channels, according to Wang et al. 2014 AMT. (Page 15, Line 6).**

**Response:** The above comment is correct. In the manuscript, we used the 200 m as the lowest altitude of the OEM-based retrieval. In the comparisons with in-situ stations, the parameterized near-surface concentrations were used. The estimation of the parameterized MAX-DOAS near-surface measurements correspond to a layer expanding from the surface to a height equal to h=dLeff(NO2)*sin(elevation angle).

For an elevation angle equal to 2° and for two characteristic horizontal sensitivities of 12 km and 8 km (Figure 8) for the Vis and UV, respectively, the altitude corresponds to 420 m and 280 m for the Vis and UV, respectively. Overall, for the Vis, the horizontal sensitivity values vary from 5 to 20 km, which corresponds to a height variation between 175 - 700 m. Similarly, for the UV, the horizontal sensitivity values vary from 5 to 13 km, which corresponds to a height variation between 175 - 450 m. The following sentence on page 15, line 24 has been modified as follows:

"**Secondly, the in-situ stations are located typically at 3-10 m altitude, while the MAX-DOAS near-surface measurements correspond to a layer extending from the surface to approximately 420 m altitude for the Vis range considering a horizontal distance of 12 km and from the surface to 280 m altitude for UV based on a horizontal distance of 8 km (see corresponding dLeff$_{NO2}$ in Fig. 8). Hence, the MAX-DOAS near-surface concentration is not fully representative of the in-situ surface concentration, as reported in Kramer et al. (2008).**"

**19. Comment: Because you mention that one azimuthal MAX-DOAS measurement samples air masses along several kilometers, what about the correlation between MAX-DOAS (geometricapproach, e.g. using 30_ measurements) and TROPOMI? (Page 16, Line 17)**

**Response:** The NO2 VCD derived by MMF in the main azimuthal direction vs TROPOMI (for the whole time period) have a correlation coefficient of 0.47 and slope equal to 0.36, while for

the NO2 VCD derived by applying the geometrical approach vs TROPOMI has a correlation coefficient of 0.47 and slope equal to 0.39. We can expect similar correlation coefficient and slope values when using the geometrical approximation since, when comparing the NO2 VCD derived by MMF with the NO2 VCD derived by the geometric approach, we find a good agreement between both datasets (R=0.95 and slope= 0.94).

**20. Comment: Actually I do not see improvement for summer. (Page 17, Line 4)**

**Response:** The term improvement is referring to the correlation coefficient values, which increases from 0.57 to 0.77 for summer. On the other hand, the slope value is almost constant for this season (it varies from 0.49 to 0.58). A summary of the regression analysis parameters derived by the three validation exercises can be found in Table 7. The corresponding paragraph has been adjusted (Page 17, line 26) as follows:
"Results displayed in Fig. 14 show that the agreement between TROPOMI and MAX-DOAS datasets is significantly improved, especially in terms of correlation (R in the 0.65-0.82 range). Owing to the improved spatial coincidence associated to the use of dual-scan MAX-DOAS data and the better spatial coincident criterion between TROPOMI and MAX-DOAS data, the scatter in the data points is also substantially reduced during all seasons and especially in winter. Another interesting feature is the improvement of the slope values, observed in all seasons (slopes in the 0.41-0.71 range). During seasons with a more homogeneous $NO_2$ field, the improvement of the slope values is less pronounced than during seasons, like winter and autumn, where the $NO_2$ field can be highly inhomogeneous."
is now:
"Results displayed in Fig. 14 show that the agreement between TROPOMI and MAX-DOAS datasets is significantly improved, especially in terms of correlation (R in the **0.60-0.85 range instead of 0.25-0.72**). Owing to the improved spatial coincidence associated to the use of dual-scan MAX-DOAS data and the better spatial coincident criterion between TROPOMI and MAX-DOAS data, the scatter in the data points is also substantially reduced during all seasons and especially in winter. Another interesting feature is the improvement of the slope values, observed **in winter and autumn (slopes of 0.81 and 0.61, respectively, instead of 0.33)**. During seasons (spring, summer) with a more homogeneous $NO_2$ field, the improvement of the slope values is less pronounced than during **seasons where the $NO_2$ field can be highly inhomogeneous (i.e. winter and autumn).** During spring, the slope value is reduced despite the better correlation. Overall, TROPOMI still underestimates MAX-DOAS measurements by about 40-50 %."

**21. Comment: Again, not clear which cloud fraction you used for cloud screening. (Page 17, Line 19)**

**Response:** See our reply to comment n°15 above.

**22. Comment: Is it mean or median? (Page 18, Line 30) Because in the conclusion you state that you are using median MAX-DOAS profiles as a priori. (Page 20, Line 17)**

**Response:** Indeed, the MAX-DOAS profiles that have been used as a priori profiles in the satellite retrieval are median MAX-DOAS profiles. This has been corrected in Page 18, Line 30. An example of a median MAX-DOAS profile during summer can be found in Figure 15.

**23. Comment: Technical corrections:**
**- associated with (Page 11, Line 29)**
**- Two modifications are introduced (Page 16, Line 24)**
**- Some words (e.g. : : : algorithm is based on NO2 : : :) are missing in the first sentence of Sect. 4.4.3 (Page 18, Line 19)**

**Response:** The three technical corrections have been included in the revised manuscript.

**Response to Anonymous Referee #2**

**1. Comment: Major drawback: Standard atmosphere**
**The authors use a standard atmosphere for their evaluation. This is quite odd, as the MAXDOAS measurements are taken right at the meteorological institute of Belgium, where I would expect that atmospheric profiles of temperature and pressure would be available on daily basis. The authors discuss the seasonality of their results. This discussion is hard to assess as the seasonality of T and p, which directly affects the $O_4$ concentration (and thus aerosol inversion and thus NO2), is just ignored. Thus the authors at least have to study the impact of the used standard atmosphere on the seasonality of results. But I would highly recommend that the authors redo the complete data analysis for real atmospheric profiles.**

**Response:** We agree with the reviewer. The standard atmosphere has been used because it was easily accessible for starting our study. Taking into account your comment, we decided to redo the complete data analysis using 20-year monthly averages of atmospheric profiles from the ECMWF Interim reanalysis extracted for Uccle, Brussels. The text in the manuscript has been adapted as follows:

"The pressure and temperature profiles are taken from the Air Force Geophysics Lab (AFGL) 1976 Standard Atmosphere (Anderson et al., 1986). The variation of the temperature profiles during one year of measurements is taken into account as an additional error on the profile retrieval."

is now:

**"The pressure and temperature profiles are prescribed using 20-year-based monthly averaged data extracted from the European Centre for Medium-Range Weather Forecasts (ECMWF) ERA Interim reanalysis (see Beirle et al., 2019) for the location of Uccle.**" (see page 5, Line 26)

and

"The near-surface VMR is then obtained by dividing the concentration of the trace gas (Eq. 4) by the air number density ($n_{air}$). For the calculation of $n_{air}$, the pressure and temperature profiles were taken from the AFGL 1976 Standard Atmosphere (Anderson et al., 1986) and are the same as used in Section 2.3.1."

is now:

"The near-surface VMR is then obtained by dividing the concentration of the trace gas (Eq. 4) by the air number density ($n_{air}$) **derived from monthly averaged temperature and pressure profiles extracted from the ERA Interim reanalysis (see Section 2.3.1)**." (see page 9, Line 24)

All the figures and discussion results have been adjusted to the new data analysis results. When comparing the $NO_2$ VCDs, $NO_2$ near-surface VMRs and MLH($NO_2$) derived using the revised atmospheric profiles with those obtained with the initial climatology, we report the following changes:

| Value | Percentage difference |
|---|---|
| **NO2 VCD Vis** | Winter: -7.1 %
Spring : -7.7 %
Summer: -8.5 %
Autumn: - 6.5 % |
| **NO2 VCD UV** | Winter: -7.2 %
Spring : -9.2 %
Summer: -10.2 %
Autumn: -8.6 % |
| **NO2 VMR Vis** | Winter: -1.8 %
Spring : -5.4 %
Summer: -9.6 %
Autumn: -5.4 % |
| **NO2 VMR UV** | Winter: -0.9 %
Spring : -5.3 %
Summer: -10.1%
Autumn: -6.1 % |

| AOD Vis | Winter: 0.6 %
Spring : -2.5 %
Summer: -6.3 %
Autumn: -5.8 % |
|---|---|
| AOD UV | Winter: 2. 9 %
Spring : -2.5 %
Summer: - 8.0 %
Autumn: -6.0 % |
| MLH (NO2) | Winter: -5.9 %
Spring : -2.7 %
Summer: 0.5 %
Autumn: -1.7 % |

The percent differences are calculated against the initial data.

**2. Comment: 5/25: For a DoF of 1, there is only 1 piece of information available (i.e. the total column). So for real "profile information" I would expect a threshold of DoF>2 rather than 1.**

**Response:** We agree with the reviewer. The main goal of this manuscript was the use of the total columns. For this reason, a DoF equal to or larger than one was used. As the complete data analysis was redone according to your first comment, a DoF equal to 2 or larger is now used in order to ensure the validity of the MAX-DOAS a-priori profiles (see page 6, Line 2). The use of a DoF equal or larger than two leads to the reduction of the number of valid MAX-DOAS scans by 4 % for the total time period examined in our study.

**3. Comment: 6/3: Is the pyrometer also pointing in the same direction as the MAX-DOAS instruments, or does it have a fixed viewing direction? If the latter, how could it provide information on cloudiness for low elevation angles?**

**Response:** The pyrometer is located on the BIRA-IASB rooftop and it is pointing at the zenith direction. The BIRA-IASB building is located 180 m away from the MAX-DOAS instrument. The cloud filtering applied in this study (only scans for a cloud percentage lower than 80 % are selected) is a first cloud filtering approach to remove scans under high cloud coverage. In the future, the approach developed in Gielen et al. (2014) will be applied, which can be used as a cloud filtering in different elevation angles (in our case, it will be applied for the elevation angle of 2°). In this approach, the color index, which is the ratio between two wavelength radiances, is used to define the sky condition (clear sky, thin clouds and high cloud coverage). First tests have already been done by using the color index in 412 nm and 500 nm as measured in the elevation angle of 2°. For high cloud coverage, 93 % of the total cases, the color index method and the pyrometer give similar result. This indicates that for the present study, pyrometer measurements can be used in first approximation for the cloud flagging of the MAX-DOAS scans

**4. Comment: 6/27: Please add a direct comparison of the NO2 results for UV and Vis and compare with the listed uncertainties.**

**Response:** A direct comparison between the NO2 near-surface VMR and NO2 VCD for UV and Vis shows that differences can reach up to 15 %. This source of uncertainty has been added to Table 4, where the total uncertainty in the MMF retrieved NO2 VMR and VCD in the Vis and UV spectral ranges is presented. As discussed into details in the manuscript (Section 4.1), the horizontal sensitivity in the Vis and UV wavelength ranges is different (larger for Vis than for UV), which results to a different horizontal air sampling.

In order to address this comment, the following sentence is added on page 6, line 32:

**"Combining all the above-mentioned sources of error, the following uncertainties for NO$_2$ retrievals are estimated: 11 % and 13 % on NO$_2$ VMR and VCD in the Vis range, respectively, and 10 % and 14 % on NO$_2$ VMR and VCD in the UV, respectively. Another source of uncertainty is estimated by comparing the NO$_2$ near-surface VMR and VCD in the UV and Vis wavelength ranges. The uncertainty is up to 15.5 % of the near-surface VMR and 15.4 % of the VCD. This percentage difference is slightly larger than the above-mentioned retrieval uncertainties. The main origin of this uncertainty is the different horizontal sensitivity in the UV and Vis wavelength ranges and therefore, a different air mass sampling."**

**5. Comment: 6/29: The subsection heading announces details on the dual scan retrieval. However, large parts of the following text just explain the par method. I propose to add a subsection dedicated to the par method for better clarity.**

**Response:** Two separate subsections were created:

'2.3 Aerosol and OEM-based profile retrievals' and '2.4 Dual-scan MAX-DOAS retrieval strategy'. In the subsection 2.4, two new subsections were created for the sake of clarity: '2.4.1 The parameterization method' and '2.4.2 Dual-scan MAX-DOAS retrieval in Uccle'.

Also small adjustments have been made to the manuscript, which can be found in Page 7, Line 6 and Page 8, Line 24.

**6. Comment: 16/4: Please put this finding in relation to other comparisons (which have also reported a low bias of TROPOMI columns) and add respective references.**

**Response:** In the present manuscript, TROPOMI tropospheric NO2 columns are found to be systematically lower than co-located MAX-DOAS measurements. This finding is in agreement with recent studies such as:

1. The study of Ialongo et al. (2020) which evaluates the TROPOMI NO2 columns in polluted conditions in Helsinki, Finland.
2. The study of Griffin et al (2019) for the Canadian Oil Sands
3. The study of Zhao et al. (2019) for urban conditions in Toronto, Canada.

A sentence is now added in Page 16, Line25 to make the link between our findings and the above references:

"**This finding is in agreement with the recent studies of Griffin et al. (2019), Zhao et al. (2019), and Ialongo et al. (2020).**"

**7. Comment: 17/26: I don't agree with the general statement that cloud effects are quasi random and do not cause systematic biases, and I don't see these bold statements supported by the presented measurements. Cloud impact is definitely "complex" and may indeed "lead to positive or negative biases". So they definitely introduce considerable scatter to the retrieved columns. But underneeth this scatter, which might look "quasi random", there are very likely systematic effects as well, which probably could only be quantified with large data samples.**

**Response:** In the present manuscript, we focus on the impact of the a-priori profile in the TROPOMI retrieval, and not on the influence of clouds. For the latter, we refer only to the existing literature. We agree with the reviewer that the cloud effects are more complex than presented here.

In TROPOMI retrieval, the FRESCO-S cloud fraction and height is used. By applying a filter in the qa_value, we exclude pixels with a cloud fraction larger than 0.5. However, pixels with small cloud fractions, which are part of the data analysis, can cause bias on the TROPOMI tropospheric NO2 columns. More precisely, over bright cities, if the TROPOMI albedo value (spatial resolution of 0.5°x0.5°) is lower than the real albedo value of the surface, the TROPOMI algorithm tends to give a non-zero cloud fraction to account for the brightness of the pixel. This can lead to an underestimation of the air mass factor (AMF) and consequently, an overestimation of the TROPOMI columns.

The following sentence has now been added in the manuscript to address the reviewer's comment on the complexity of cloud characterization and filtering in satellite data (see Page 18, Section 4.4.1):

"**This cloud information is used as an input in a cloud-correction scheme applied to NO$_2$ retrieval (van Geffen et al., 2019). Cloud-induced errors are complex and can lead to**

**positive or negative biases on the tropospheric NO₂ column resulting to considerable scatter to the retrieved columns, especially for small cloud fractions."**

**8. Comment: Table 1: Please add the different sets of azimuth angles to Fig. 1, color coded for the three time periods.**

5 **Response:** Figure 1 has been modified and different sets of azimuth angles corresponding to the three time periods are now shown with different colors:

[Figure]

**9. Comment: Table 4: Please add results from a direct comparison between NO2 UV and**
10 **vis - is the difference within the listed total uncertainty range?**

**Response:** As mentioned in the manuscript (see Page 6, Line 34), the difference between NO2 Vis and UV is about 15.5 % in the NO2 near-surface VMR and 15.4 % in the NO2 tropospheric VCD. This difference is slightly larger than the listed total uncertainty range.
15 These values have been added in Table 4.

**10. Comment: Fig. 1: Please add the different sets of azimuth angles, color coded for the three time periods. What is the meaning of the symbols on top of the azimuth direction lines?**

**Response:** Figure 1 has been updated according to your recommendation. The meaning of the symbols on top of the azimuth direction lines showed the direction (from the MAX-DOAS instrument to the maximum horizontal distance). For the simplicity of the figure, the symbols were excluded and simple lines were used.

**11. Comment: Fig. 7: The landscape background is not helpful here, but rather disturbing. I propose to show no background, but instead real color coded VCDs for both TROPOMI and MAXDOAS.**

30 **Response:** Figure 7 has been updated according to your recommendation:

[Figure]

**12. Comment: Fig. 12: Several MAX-DOAS VCDs have no errorbar. I assume that for these days no std could be calculated. I propose to also show the mean/typical uncertainty of single VCDs derived from the MAX-DOAS inversion, which would probably be more consistent for the time period. This might be added as second error bar with e.g. light color.**

**Response:** The MAX-DOAS VCDs that have no error bar correspond to days where no standard deviation (std) can be calculated. Following reviewer's comment, we have added the typical MAX-DOAS inversion uncertainty (reported in Table 4) on each data point with grey light color.

**13. Comment: Fig. 15/16: Please add the corresponding a-priori profiles used in the TROPOMI retrieval, and add a discussion of this comparison.**

**Response:** The corresponding a-priori profiles used in the TROPOMI retrieval have been added to Figure 15 and 16:

Figure 15

[Figure]

Figure 16

[Figure]

In order to address this comment, the following sentences are added:

**"Figures 15 and 16 present a-priori NO₂ profiles used in the TROPOMI retrieval. One can see that the a-priori TROPOMI NO₂ profiles are lower than the median MAX-DOAS NO₂ profile (except for the example summer day in Fig. 16). In the first kilometers above the surface, the difference between both profiles is about 35 % in Fig. 15 and can be up to**
10 **80 % for the example autumn day in Fig. 16. "** (Page 19, Line 24)

**14. Comment: Minor comments:**
**1/18: "concentrations" should be "volume mixing ratios"**
**3/26: "eventual" should be "potential" or "possible"**
**6/2: Skip the comma between "measurements" and "strongly"**
**6/10: "Smoothing error"**
**6/30: Please define what exactly is meant by "near surface VMR"**
**8/29: "are equal" should be "are similar"**
**9/21: "close to zero" is not unphysical**

**Response:** The above technical corrections have been implemented in the revised manuscript.

[revised manuscript text omitted]

---

## Author Response (AR2)

**Response to the Associate Editor**

We would like to thank Lok Lamsal for the thoughtful examination of our manuscript. Please find below our responses to each comment:

1. Section 3, Page 33, lines 4-7: "The AMF look-up tables are calculated on a 1o x 1 o latitude-longitude grid using NO2 vertical profiles from the TM5-MP model (Williams et al., 2017)". This is incorrect. The AMFs are calculated for each individual field of view (FOV) or pixel. However, NO2 profiles used in the retrievals are taken from TM5 model at 1 deg x 1 deg resolution. Please, correct the statement.

   **Response:** The statement in Section 3, Page 12, lines 27-29 is now corrected as follows:

   **The AMFs are calculated for each individual field of view (FOV) or pixel by using a-priori information about the viewing and solar geometry, surface pressure and NO$_2$ profile shape from the 1$^o$ x 1$^o$ TM5-MP model (Williams et al., 2017), 0.5$^o$ x 0.5$^o$ surface albedo climatology, cloud fraction and cloud height.**

2. Section 4.2, page 36, lines 5-9: You described two reasons for the observed difference between MAX-DOAS and surface NO2 data. The discrepancies could also arise from errors in either MAX-DOAS retrievals or surface NO2 data. I could not locate instrument type(s) used at the surface sites. Some commercial NO2 monitors have biases that are well documented.

   **Response:** We agree with the Editor. Unfortunately, information about the in-situ instrument type could not be found. The following sentence is now added in the manuscript (Page 15, Lines 28-30):

   **Thirdly, the comparison discrepancies could also come from errors in either MAX-DOAS retrievals, as documented in Table 4 & 5, or surface NO$_2$ measurements because of instrumental errors and biases.**

3. Section 4.4.2, page 39, lines 22-23 : "…..the surface albedo values used in the TROPOMI retrieval have a spatial resolution of 13 x 24 km2 (OMI spatial resolution). In reality, inside an area of 13 x 24 km2 in an urban environment…". This is not correct. The spatial resolution of surface reflectivity used in the TROPOMI algorithm is 0.5 deg x 0.5 deg from Kleipool et al (2008). It is a monthly mean climatology. Please, correct the statement. You may also want to cite other recent works on the impact of surface reflectivity on NO2 retrievals (e.g., GOME-2 NO2 DLR and OMI NO2 from NASA).

   **Response:** The statement in Section 4.4.2, Page 19, lines 4-6 is now corrected as follows:

   **However, we should keep in mind that the surface albedo values used in the TROPOMI NO$_2$ AMF calculations have a spatial resolution of 0.5$^o$ x 0.5$^o$ and are monthly mean climatology values (Kleipool et al., 2008). In reality, inside an area 0.5$^o$ x 0.5$^o$ of an urban environment, …**

   The reference of Kleipool et al. (2008) is now added in the Reference list.

   Additionally, we have added the two suggested recent studies as follows (Page 19, lines 9-10):

[revised manuscript text omitted]